# Adaptive exponential weighted composite sliding mode-based direct yaw moment control for four-wheel independently actuated autonomous vehicles

Zhengyong Tao[1], Mingming Wu[2‡], Min Qu[1‡], Hui Wu[1‡], Banglai Sun[2‡], Deqiang Xie[1‡], Zhongzhi Tong[3]*

1 School of Automotive Engineering, Wuhu University, Wuhu, Anhui, China, 2 School of Intelligent Manufacturing, Wuhu University, Wuhu, Anhui, China, 3 School of Mechanical Engineering, Nanjing University of Science and Technology, Nanjing, Jiangsu, China

☯ These authors contributed equally to this work.
‡ MW, MQ, HW, BS and DX also contributed equally to this work.
* tzz01591088@126.com

## Abstract

This paper introduces a novel Direct Yaw Moment Control (DYC) scheme for autonomous vehicles with Four-Wheel Independent Actuation (FWIA). In the upper-layer control strategy, an Adaptive Exponential Weighted Composite Sliding Mode Controller (AEWC-SMC) is proposed by incorporating a nonlinear weighting factor into the sliding mode surface and designing a composite reaching law. For the lower-layer torque allocation, a Dynamic Weight Minimum Energy Allocation (DWMEA) method is developed, which achieves optimal four-wheel torque distribution without iterative computation. By formulating the objective function to minimize weighted energy consumption and introducing adaptive dynamic weight parameters that account for vertical load, steering angle, and vehicle speed, this method adaptively realizes optimal torque allocation. MATLAB/Simulink simulation results demonstrate that compared with traditional Sliding Mode Control (SMC) schemes, the proposed control strategy exhibits higher tracking accuracy, faster convergence speed, and enhanced handling stability.

## Introduction

To address global climate change and advance strategic initiatives for green transition, the development of New Energy Vehicles (NEVs) has accelerated rapidly. In this context, Four-Wheel Independent Drive (4WID) electric vehicles have emerged as a new research frontier [1,2], demonstrating significant advantages in response efficiency and control flexibility. Consequently, autonomous vehicles based on 4WID electric vehicle chassis designs are poised to become a major developmental direction. The core research focus now centers on ensuring driving reliability, stability, and safety under extreme operating conditions [3–5].

**Data availability statement:** All relevant data are within the manuscript and its Supporting Information files.

**Funding:** This work was supported in part by the 2024 Anhui Provincial Natural Science Key Research Project (Grant No. 2024AH052007) and in part by the 2024 Institutional Scientific Research Project (Grant No. WHKY-202413). The funding (total 103,000 CNY) was awarded to author Z.Y.T. (Zhengyong Tao). The sponsors provided financial support but had no role in study design, data collection/analysis, decision to publish, or preparation of the manuscript.

**Competing interests:** The authors have declared that no competing interests exist.

Extensive research has demonstrated that Direct Yaw Moment Control (DYC) stands as one of the most effective control strategies for ensuring superior handling stability during extreme operating conditions [6–9]. The DYC framework predominantly employs a hierarchical architecture: the upper-layer controller module generates the external yaw moment output, while the lower-layer allocation design module optimally distributes this moment to the four wheels, thereby significantly enhancing vehicle handling stability performance [10]. Notably, Fengxi Xie et al. [11] recently proposed an adaptive sliding mode trajectory tracking controller, which improves path tracking accuracy through vector field guidance law and intelligent optimization algorithms. However, this study primarily addresses the trajectory tracking problem for centrally driven vehicles, whereas the DYC scheme focuses on collaborative control for yaw stability and sideslip safety for four-wheel independently driven vehicles under extreme operating conditions, demonstrating complementary relationships in control objectives and application scenarios. In light of the outstanding contributions of Fengxi Xie et al.'s research in the field of path tracking, this paper adopts a layered DYC framework to address the uncovered stability issues under extreme operating conditions.

Current research on upper-layer controller design primarily encompasses PID control [12], $H\infty$ control [13,14], Model Predictive Control (MPC) [15–17] and Sliding Mode Control (SMC) [18,19]. Among these, SMC has been widely adopted in DYC up-per-layer controller design due to its inherent advantages of strong robustness, rapid response characteristics, and insensitivity to parameter variations and external disturbances. Hasan Alipour et al. [20] proposed a novel modified integral sliding mode controller by introducing a proportional-integral term and an online parameter optimization algorithm, achieving effective control over the lateral stability of four-wheel independently driven electric vehicles. Their simulations demonstrated that the proposed controller achieves faster response and better stability compared to traditional Sliding Mode Control (SMC). However, this method suffers from issues of high-dimensional parameter tuning and heavy reliance on the accuracy of the vehicle dynamics model. Xiaoyu Li et al. [21] proposed a stability index-based adaptive sliding mode control scheme, which introduced a quantitative stability index by constructing a three-zone stability boundary (stable zone radius, transition zone radius, unstable zone radius) on the phase plane of front/rear tire slip angles. Simulation results also demonstrated that this scheme can effectively improve the handling stability of the vehicle. However, the phase plane analysis method employed in the study relies on quasi-steady-state assumptions, and its stability discrimination capability under extreme operating conditions requires further investigation. Houzhong Zhang et al. [22] proposed a Fuzzy Sliding Mode Controller (FSMC). They designed FSMC as the core decision-making layer of the control method to calculate the required additional yaw moment based on estimated sideslip angle. Simulation results indicated that FSMC exhibits strong chattering suppression capability and effectively enhances the system's manipulation stability. However, this method suffers from inadequate coverage under complex operating conditions, and its adaptive adjustment capability of fuzzy rules may deteriorate due to intensified tire nonlinear

characteristics. Xiaoqiang Sun et al. [23] proposed a Nonsingular Terminal Sliding Mode (NTSM) control method. Simulation results validated the effectiveness of this controller in terms of lateral stability and tracking accuracy for path following under four extreme conditions. However, the allocation of external yaw moment in this study involves adjustments of weighting coefficients, which are obtained through iterative computation. This approach may suffer from insufficient real-time performance under extreme operating conditions. Existing research has made remarkable progress in mitigating chattering and enhancing robustness. Nevertheless, aspects such as adaptability to complex operating conditions and real-time computational efficiency must still be considered. Therefore, significant research space remains in sliding mode control design.

The lower-layer module of DYC systems is responsible for allocating the external yaw moment to four-wheel torque. Yong Chen et al. [24] proposed dynamic torque allocation using the weighted least squares method, with simulations verifying the effectiveness of the allocation algorithm. However, the torque distribution process requires iterative computation, which may suffer from real-time performance issues under extreme operating conditions. Xiao Hu et al. [25] proposed an Energy-Saving Distribution (ESD) strategy suitable for extreme operating conditions. Through a dual-stage distribution considering energy-saving performance and stability performance separately, it achieves energy conservation and stability enhancement without mutual interference. However, the practical allocation design process still involves iterative computation issues, which requires further optimization. Hao Cui et al. [26] proposed a quadratic programming-based optimal allocation strategy grounded in sliding mode control, achieving optimal four-wheel torque distribution and effectively improving the vehicle's handling stability. However, it similarly involves iterative computation issues. The aforementioned studies all successfully accomplish optimal torque allocation, yet uniformly rely on obtaining optimal values through iterative computation. Insufficient iteration counts may fail to secure optimal values, while excessive iterations prolong control response times and cause deterioration of control effectiveness. Therefore, developing a real-time efficient allocation algorithm that eliminates iterative computation remains critically important in lower-layer allocation design.

To address this, the present study proposes a novel DYC system architecture with two primary contributions: 1) An Adaptive Exponential Weighted Composite Sliding Mode Controller (AEWC-SMC) is designed for the upper-layer control strategy. By introducing a nonlinear weighting factor into the sliding mode surface, this controller achieves adaptive precision control through rapid response to large sideslip angle deviations and linear control for minor deviations. Furthermore, a composite reaching law incorporating linear terms, smooth nonlinear terms, and fractional-order nonlinear terms is developed, enabling swift correction under large errors and smooth transition during small errors to achieve rapid convergence and chattering suppression. 2) A Dynamic Weight Minimum Energy Allocation (DWMEA) method requiring no iterative computation is proposed for the lower-layer allocation design. By formulating an objective function to minimize weighted energy consumption and introducing adaptive dynamic weight parameters that account for vertical load, steering angle, and vehicle speed, this method adaptively realizes optimal four-wheel torque distribution.

The remainder of this paper is organized as follows. Section II introduces the 7-DOF and 2-DOF vehicle models. Section III details the design of the upper-layer AEWC-SMC. Section IV presents the DWMEA method for lower-layer allocation de-sign. Section V describes comparative simulation experiments under two operational conditions. Section VI provides concluding remarks.

## Vehicle model and analysis

### 7-DOF vehicle dynamics mode

To accurately characterize vehicle motion characteristics under diverse operating conditions, this study establishes a 7-Degree-of-Freedom (7-DOF) vehicle dynamics model [27] that comprehensively incorporates nonlinear dynamic characteristics. The model integrates three degrees of freedom of body motion (longitudinal, lateral and yaw) with four degrees of freedom of wheel rotation, as shown in Fig 1. Through coupling spatial body motion and wheel dynamic characteristics, it effectively captures vehicle dynamic responses in complex operating scenarios.

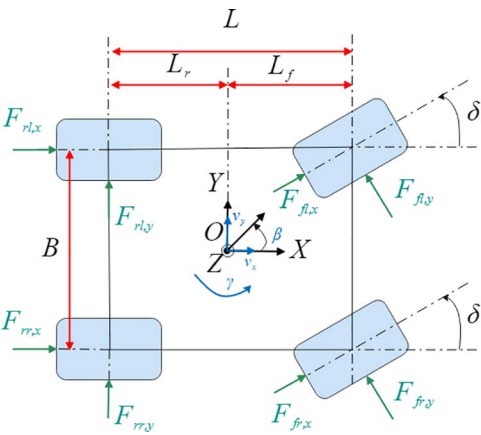

**Fig 1. Top view of the 7-DOF vehicle model.**

The vehicle dynamics model can be expressed as

$$m\left(\dot{v}_x - v_y\gamma\right) = \left(F_{fl,x} + F_{fr,x}\right)\cos\delta - \left(F_{fl,y} + F_{fr,y}\right)\sin\delta \\ + F_{rl,x} + F_{rr,x}$$

(1)

$$m\left(\dot{v}_y + v_x\gamma\right) = \left(F_{fl,x} + F_{fr,x}\right)\sin\delta + \left(F_{fl,y} + F_{fr,y}\right)\cos\delta \\ + F_{rl,y} + F_{rr,y}$$

(2)

$$I_z\dot{\gamma} = \left[\left(F_{fl,y} + F_{fr,y}\right)\cos\delta\right]L_f - \left(F_{rl,y} + F_{rr,y}\right)L_r + \frac{B}{2}\left(F_{fl,y} - F_{fr,y}\right)\sin\delta \\ + \left[\left(F_{fl,x} + F_{fr,x}\right)\sin\delta\right]L_f + \frac{B}{2}\left[\left(F_{fr,x} - F_{fl,x}\right)\cos\delta + F_{rr,x} - F_{rl,x}\right]$$

(3)

$$J_i\dot{\omega}_i = T_i - F_{i,x}R, i = fl, fr, rl, rr$$

(4)

Where $B$ is the vehicle track width; $L_f$ is the distance from the center of mass to the front axle; $L_r$ is the distance from the center of mass to the rear axle; $R$ represents the tire radius; $m$ is the vehicle mass; $I_z$ is the yaw moment of inertia; $\delta$ is the front wheel steering angle; $v_x$ is the longitudinal velocity; $v_y$ is the lateral velocity; $\beta$ is the sideslip angle; $\gamma$ is the yaw rate; $F_{fl,x}$, $F_{fr,x}$, $F_{rl,x}$ and $F_{rr,x}$ are the longitudinal forces of the front-left, front-right, rear-left, and rear-right tires, respectively; $F_{fl,y}$, $F_{fr,y}$, $F_{rl,y}$ and $F_{rr,y}$ are the lateral forces of the front-left, front-right, rear-left, and rear-right tires, respectively; $J_i$ is the wheel rotational inertia; $\omega_i$ are the angular velocities; $T_i$ are the driving or braking torques applied to the corresponding tires.

The vertical tire loads can be calculated by:

$$\begin{cases} F_{fl,z} = m_wg + \frac{m}{L}\left(\frac{1}{2}gL_r - \frac{1}{2}a_xh_g - \frac{L_r}{B}a_yh_g\right) \\ F_{fr,z} = m_wg + \frac{m}{L}\left(\frac{1}{2}gL_r - \frac{1}{2}a_xh_g + \frac{L_r}{B}a_yh_g\right) \\ F_{rl,z} = m_wg + \frac{m}{L}\left(\frac{1}{2}gL_f + \frac{1}{2}a_xh_g - \frac{L_f}{B}a_yh_g\right) \\ F_{rr,z} = m_wg + \frac{m}{L}\left(\frac{1}{2}gL_f + \frac{1}{2}a_xh_g + \frac{L_f}{B}a_yh_g\right) \end{cases}$$

(5)

where $m_w$ is the tire mass; $g$ is the gravitational acceleration; $h_g$ is the vehicle center of gravity height; $a_x$ and $a_y$ are the longitudinal acceleration and lateral acceleration, respectively.

The longitudinal tire forces $F_{i,x}$, ($i = fl, fr, rl, rr$) and lateral forces $F_{i,y}$ are coupled through the friction circle constraint $\sqrt{F_{i,x}^2 + F_{i,y}^2} \leq \mu F_{i,z}$, whose nonlinear interaction directly affects stability. When a tire simultaneously experiences longitudinal slip and sideslip angle, the resultant force is constrained by the friction circle boundary. If $|F_{i,x}|$ increases (e.g., during emergency acceleration), $F_{i,y}$ must decrease to maintain $\sqrt{F_{i,x}^2 + F_{i,y}^2} \leq \mu F_{i,z}$, thereby weakening steering response. Under extreme operating conditions (e.g., steering at high speed with low adhesion coefficient), the friction circle constraint is easily violated, potentially leading to tire saturation and yaw instability, i.e., posing a stability risk. Simultaneously, the dynamic variation of vertical load $F_{i,z}$ further influences the friction circle radius $\mu F_{i,z}$.

**Tire model**

The mechanical characteristics of tires directly influence vehicle handling stability, making the establishment of an accurate tire model essential for vehicle dynamics analysis. The Magic Formula (MF) tire model, a widely adopted semi-empirical model in engineering applications, provides precise characterization of nonlinear mechanical responses in the tire contact patch. Validated through decades of real-vehicle testing, this model has matured into an established engineering application framework. Considering both model accuracy and practical engineering utility, this study employs the MF tire model. The MF formulation is expressed as

$$y(x) = D \sin\left(C \arctan\left(Bx - E\left(Bx - \arctan\left(Bx\right)\right)\right)\right) \tag{6}$$

where $y$ represents the longitudinal or lateral tire force; $x$ denotes the tire slip ratio or sideslip angle; $B$, $C$, $D$ and $E$ are the stiffness factor, shape factor, peak factor, and curvature factor, respectively.

The nominal vertical load increment is defined as

$$df_z = \frac{F_z - F_{z0}}{F_{z0}} \tag{7}$$

where $F_z$ is the tire vertical load; $F_{z0}$ is the nominal vertical load.

Under pure slip conditions, the longitudinal force based on the MF model is expressed as

$$\begin{cases} F_{x0} = D_x \sin\left(C_x \arctan\left(B_x \lambda_x - E_x\left(B_x \lambda_x - \arctan\left(B_x \lambda_x\right)\right)\right)\right) + S_{Vx} \\ \lambda_x = \lambda_i + S_{Hx} \end{cases} \tag{8}$$

where

$$\begin{cases} C_x = x_1 \\ D_x = \left(x_2 + x_3 df_z\right)\left(1 - x_4 \theta_\gamma^2\right) \\ E_x = \left(x_5 + x_6 df_z + x_7 df_z^2\right)\left(1 - x_8 \mathrm{sgn}\left(\lambda_x\right)\right) \\ B_x = \frac{F_z\left(x_9 + x_{10} df_z\right) e^{x_{11} F_z}}{C_x D_x} \\ S_{Vx} = F_z\left(x_{12} + x_{13} df_z\right) \\ S_{Hx} = x_{14} + x_{15} df_z \end{cases} \tag{9}$$

where $x_i$ ($i = 1, 2, \cdots, 15$) are fitting parameters; $\theta_\gamma$ is the camber angle; $\lambda_i$ ($i = fl, fr, rl, rr$) is the longitudinal slip ratio, calculated by:

$$\lambda_i = \begin{cases} \frac{\omega_i R - v_i}{\omega_i R} \left( v_i < \omega_i R, \omega_i \neq 0 \right), acceleration \\ \frac{v_i - \omega_i R}{v_i} \left( v_i \geq \omega_i R, v_i \neq 0 \right), braking \end{cases} \tag{10}$$

where $v_i \, (i = fl, fr, rl, rr)$ is the wheel center velocity, derived as

$$\begin{cases} v_{fl} = \left( v_x - \frac{B}{2}\gamma \right) \cos \delta + \left( v_y + L_f \gamma \right) \sin \delta \\ v_{fr} = \left( v_x + \frac{B}{2}\gamma \right) \cos \delta + \left( v_y + L_f \gamma \right) \sin \delta \\ v_{rl} = v_x - \frac{B}{2}\gamma \\ v_{rr} = v_x + \frac{B}{2}\gamma \end{cases} \tag{11}$$

For pure sideslip conditions, the lateral tire force based on the MF model is given by:

$$\begin{cases} F_{y0} = D_y \sin \left( C_y \arctan \left( B_y \alpha_i - E_y \left( B_y \alpha_y - \arctan \left( B_y \alpha_y \right) \right) \right) \right) + S_{Vy} \\ \alpha_y = \alpha_i + S_{Hy} \end{cases} \tag{12}$$

where

$$\begin{cases} C_y = y_1 \\ D_y = \mu_y F_z, \mu_y = \left( y_2 + y_3 df_z \right) \left( 1 - y_4 \theta_\gamma^2 \right) \\ E_y = \left( y_5 + y_6 df_z \right) \left[ 1 - \left( y_7 + y_8 \theta_\gamma \right) \text{sgn} \left( \alpha_y \right) \right] \\ B_y = y_9 F_{z0} \sin \left[ 2 \arctan \left( \frac{F_z}{y_{10} F_{z0}} \right) \right] \left( 1 - y_{11} \left| \theta_\gamma \right| \right) / \left( C_y D_y \right) \\ S_{Vy} = F_z \left[ y_{12} + y_{13} df_z + \left( y_{14} + y_{15} df_z \right) \theta_\gamma \right] \\ S_{Hy} = y_{16} + y_{17} df_z + y_{18} \theta_\gamma \end{cases} \tag{13}$$

where $y_i \, (i = 1, 2, \cdots, 18)$ represent fitting parameters; $\alpha_i \, (i = fl, fr, rl, rr)$ is the tire sideslip angle, defined as

$$\begin{cases} \alpha_{fl} = - \left( \delta - \arctan \left( \frac{v_y + L_f \gamma}{v_x - B\gamma/2} \right) \right) \\ \alpha_{fr} = - \left( \delta - \arctan \left( \frac{v_y + L_f \gamma}{v_x + B\gamma/2} \right) \right) \\ \alpha_{rl} = \arctan \left( \frac{v_y - L_r \gamma}{v_x - B\gamma/2} \right) \\ \alpha_{rr} = \arctan \left( \frac{v_y - L_r \gamma}{v_x + B\gamma/2} \right) \end{cases} \tag{14}$$

The preceding formulations primarily describe tire mechanical equations under independent longitudinal slip or pure sideslip conditions. However, during actual vehicle operation, longitudinal and lateral forces often interact synergistically, leading to significant deviations between theoretical predictions and actual mechanical responses under coupled conditions. Consequently, the following modifications are applied to the tire model.

The longitudinal force equation under coupled slip conditions is modified as

$$F_x = F_{x0} \cdot \psi_x \tag{15}$$

where $\psi_x$ denotes the weighting factor characterizing the tire force coupling effect, expressed by:

$$\begin{cases} \psi_x = \dfrac{\cos\left\{C_{x\alpha} \arctan\left[B_{x\alpha}\alpha_s - E_{x\alpha}\left(B_{x\alpha}\alpha_s - \arctan\left(B_{x\alpha}\alpha_s\right)\right)\right]\right\}}{\cos\left\{C_{x\alpha} \arctan\left[B_{x\alpha}S_{Hx\alpha} - E_{x\alpha}\left(B_{x\alpha}S_{Hx\alpha} - \arctan\left(B_{x\alpha}S_{Hx\alpha}\right)\right)\right]\right\}} \\ \alpha_s = \alpha_i + S_{Hx\alpha} \end{cases} \tag{16}$$

where

$$\begin{cases} B_{x\alpha} = a_1 \cos\left[\arctan\left(a_2\lambda_i\right)\right] \\ C_{x\alpha} = a_3 \\ E_{x\alpha} = a_4 + a_5 df_z \\ S_{Hx\alpha} = a_6 \end{cases} \tag{17}$$

where $a_i \, (i = 1, 2, \cdots, 6)$ are fitting parameters.

The lateral force equation under coupled slip conditions is adjusted to:

$$F_y = F_{y0} \cdot \psi_y + S_{Vy\lambda} \tag{18}$$

where $\psi_y$ denotes the weighting factor characterizing the tire force coupling effect, defined as

$$\begin{cases} \psi_y = \dfrac{\cos\left\{C_{y\lambda} \arctan\left[B_{y\lambda}\lambda_s - E_{y\lambda}\left(B_{y\lambda}\lambda_s - \arctan\left(B_{y\lambda}\lambda_s\right)\right)\right]\right\}}{\cos\left\{C_{y\lambda} \arctan\left[B_{y\lambda}S_{Hy\lambda} - E_{y\lambda}\left(B_{x\alpha}S_{Hy\lambda} - \arctan\left(B_{y\lambda}S_{Hy\lambda}\right)\right)\right]\right\}} \\ \lambda_s = \lambda_i + S_{Hy\lambda} \end{cases} \tag{19}$$

where

$$\begin{cases} B_{y\lambda} = b_1 \cos\left\{\arctan\left[b_2\left(\alpha_i - b_3\right)\right]\right\} \\ C_{y\lambda} = b_4 \\ E_{y\lambda} = b_5 + b_6 df_z \\ S_{Hy\lambda} = b_7 + b_8 df_z \\ S_{Vy\lambda} = \mu_y F_z\left(b_9 + b_{10} df_z + b_{11}\theta_\gamma\right)\cos\left(\arctan\left(b_{12}\alpha_i\right)\right)\sin\left(b_{13}\arctan\left(b_{14}\lambda_i\right)\right) \end{cases} \tag{20}$$

where $b_i \, (i = 1, 2, \cdots, 14)$ are fitting parameters.

**Reference model**

This study employs a linear 2-Degree-of-Freedom (2-DOF) vehicle model to calculate the desired yaw rate and desired sideslip angle. As illustrated in Fig 2, the model focuses on lateral and yaw motions while incorporating simplifying assumptions—neglecting suspension roll, tire nonlinear slip characteristics, and vertical load transfer—to enable effective analysis of vehicle dynamic behavior.

The 2-DOF dynamic model [28] is governed by the following equations:

$$\begin{cases} mv_x\left(\dot{\beta} + \gamma\right) = \left(k_f + k_r\right)\beta + \dfrac{\left(L_f k_f - L_r k_r\right)\gamma}{v_x} - k_f\delta \\ I_z\dot{\gamma} = \left(L_f k_f - L_r k_r\right)\beta + \dfrac{\left(L_f^2 k_f + L_r^2 k_r\right)\gamma}{v_x} - L_f k_f\delta \end{cases} \tag{21}$$

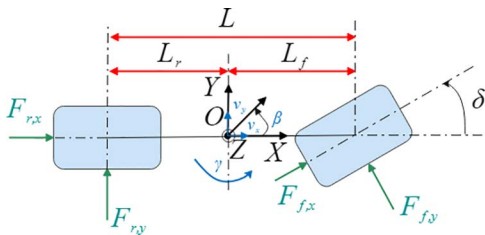

**Fig 2. 2-DOF vehicle dynamics model.**

where $k_f$ and $k_r$ represent the equivalent cornering stiffness of the front and rear axles, respectively.

By combining Equation (21) with the friction constraint boundary analysis method from references [29,30], and introducing dynamic stability thresholds for yaw rate and sideslip angle, the theoretical reference values $\gamma_{ref}$ and $\beta_{ref}$ are derived. These reference quantities serve as input targets for the upper-layer controller, formulated as

$$\begin{cases} \gamma_{ref} = \min\left\{\left|\frac{v_x\delta}{L(1+Kv_x^2)}\right|, \left|\frac{0.85\mu g}{v_x}\right|\right\} \cdot \text{sgn}(\delta) \\ \beta_{ref} = \min\left\{\left|\frac{(L_rLk_f+mL_fv_x^2)\delta}{L^2k_f(1+Kv_x^2)}\right|, \left|\mu g\left(\frac{L_r}{v_x^2} + \frac{mL_f}{k_rL}\right)\right|\right\} \cdot \text{sgn}(\delta) \\ K = \frac{m}{L^2}\left(\frac{L_f}{k_r} - \frac{L_r}{k_f}\right) \end{cases} \tag{22}$$

## Upper controller design

The primary objectives of the vehicle's DYC system are: 1) The upper-layer controller must calculate the required yaw moment accurately in real-time under extreme conditions, and 2) The lower-layer allocation mechanism must optimally distribute this yaw moment to the four wheels. To achieve this, the AEWC-SMC is designed for the upper-layer strategy, and the DWMEA method is proposed for the lower-layer allocation. The overall DYC architecture is shown in Fig 3.

The design of AEWC-SMC in this paper fundamentally differs from traditional composite sliding mode control in two aspects: 1. The nonlinear weighting factor achieves adaptive error sensitivity, rapidly amplifying control authority under large sideslip deviations while maintaining linear accuracy during small deviations, a characteristic absent in traditional composite sliding mode control. 2. The composite reaching law innovatively integrates fractional-order nonlinearity with steering state-dependent exponential selection, a feature absent in traditional composite sliding mode control. The design of the AEWC-SMC is detailed as follows.

Based on Equation (21) and considering the yaw moment effect, the lateral 2-DOF vehicle model is expressed as [31]:

$$\begin{cases} \dot{\beta} = \frac{k_f+k_r}{mv_x}\beta + \left(\frac{L_fk_f-L_rk_r}{mv_x^2} - 1\right)\gamma - \frac{k_f}{mv_x}\delta \\ \dot{\gamma} = \frac{L_fk_f-L_rk_r}{I_z}\beta + \frac{L_f^2k_f+L_r^2k_r}{I_zv_x}\gamma - \frac{L_fk_f}{I_z}\delta + \frac{\Delta M_z}{I_z} \end{cases} \tag{23}$$

where $\Delta M_z$ denotes the additional external yaw moment.

Define the sideslip angle error and yaw rate error:

$$e_\beta = \beta_{ref} - \beta, e_\gamma = \gamma_{ref} - \gamma \tag{24}$$

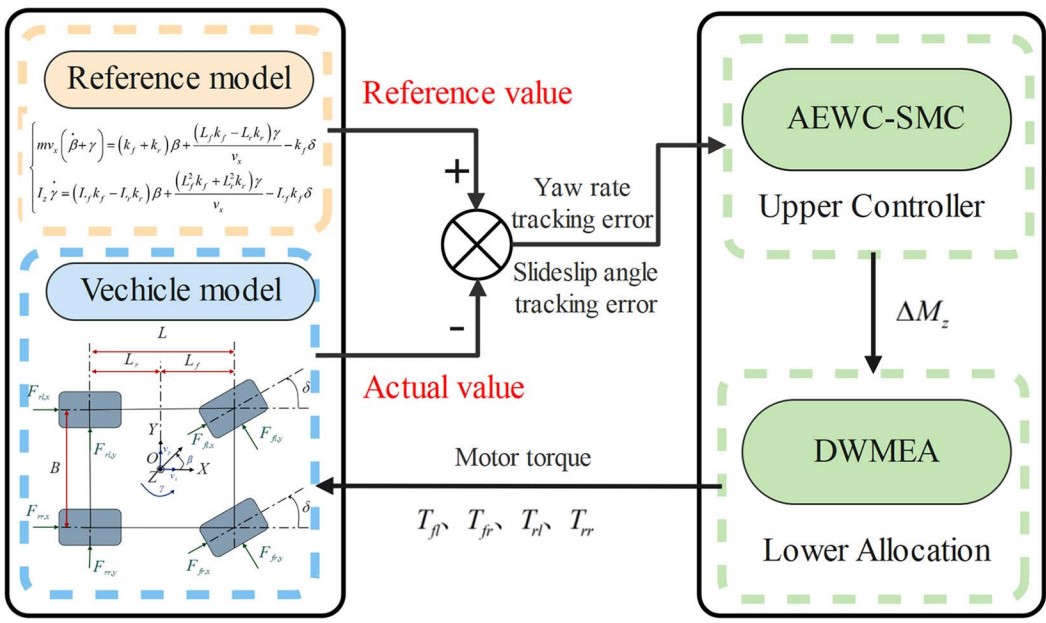

**Fig 3. Control framework illustration.**

To enhance control authority against large sideslip disturbances, a nonlinear weighting factor $e^{\kappa e_\beta^2}$ is introduced during the sliding surface construction. The designed sliding surface function is expressed as

$$s = e_\gamma + \lambda e^{\kappa e_\beta^2} \cdot e_\beta \tag{25}$$

where $\lambda > 0$ is the weighting adjustment parameter controlling the baseline weight of the sideslip angle error, and $\kappa > 0$ is the exponential factor. When the sideslip angle error is large, the nonlinear weighting factor $e^{\kappa e_\beta^2}$ provides stronger control response to rapidly reduce the error impact. When the sideslip angle error is small, $e^{\kappa e_\beta^2} \approx 1$, causing the sliding surface to approximate a conventional linear form, thereby achieving precise control.

To achieve rapid correction under large errors and smooth transition under small errors, a composite reaching law integrating linear, smooth nonlinear, and fractional-order nonlinear terms is designed:

$$\dot{s} = -\alpha s - a_1 \tanh\frac{s}{\varepsilon} - a_2 sign(s)\,|s|^\tau \tag{26}$$

where $\alpha > 0$ is the convergence gain, $a_1 > 0$ and $a_2 > 0$ are the gains for the smooth switching term and nonlinear acceleration term, $\varepsilon > 0$ is the boundary layer thickness, and $0 < \tau < 1$ is the fractional exponent.

By differentiating the sliding surface function $s$ using Equation (23):

$$\begin{aligned}
\dot{s} &= (\dot{\gamma}_{ref} - \dot{\gamma}) + \lambda e^{\kappa e_\beta^2} \cdot \left(\dot{\beta}_{ref} - \dot{\beta}\right) + \lambda e_\beta \cdot e^{\kappa e_\beta^2} \cdot 2\kappa e_\beta \cdot \left(\dot{\beta}_{ref} - \dot{\beta}\right) \\
&= (\dot{\gamma}_{ref} - \dot{\gamma}) + \lambda e^{\kappa e_\beta^2}\left(1 + 2\kappa e_\beta^2\right)\left(\dot{\beta}_{ref} - \dot{\beta}\right) \\
&= \left(\frac{L_f k_f - L_r k_r}{I_z}e_\beta + \frac{L_f^2 k_f + L_r^2 k_r}{I_z}e_\gamma - \frac{\Delta M_z}{I_z}\right) + \\
&\quad \lambda e^{\kappa e_\beta^2}\left(1 + 2\kappa e_\beta^2\right)\left[\frac{k_f + k_r}{mv_x}e_\beta + \left(\frac{L_f k_f - L_r k_r}{mv_x^2} - 1\right)e_\gamma\right]
\end{aligned} \tag{27}$$

Combining Equations (26) and (27), the external yaw moment is derived as

$$\Delta M_z = I_z \cdot \begin{Bmatrix} \left[ \alpha s + a_1 \tanh \frac{s}{\varepsilon} + a_2 sign\,(s)\,|s|^\tau \right] + \\ \left[ \frac{L_f k_f - L_r k_r}{I_z} e_\beta + \frac{L_f^2 k_f + L_r^2 k_r}{I_z v_x} e_\gamma \right] + \\ \lambda e^{\kappa e_\beta^2} \left( 1 + 2\kappa e_\beta^2 \right) \left[ \frac{k_f + k_r}{m v_x} e_\beta + \left( \frac{L_f k_f - L_r k_r}{m v_x^2} - 1 \right) e_\gamma \right] \end{Bmatrix} \qquad (28)$$

To validate the stability of the designed AEWC-SMC system, Lyapunov theory is employed for stability analysis.

Proof: Define the Lyapunov function:

$$V = \frac{1}{2} s^2 \qquad (29)$$

Differentiating $V$ with respect to time using Equation (26):

$$\begin{aligned} \dot{V} = s\dot{s} &= s \left[ -\alpha s - a_1 \tanh \frac{s}{\varepsilon} - a_2 sign\,(s)\,|s|^\tau \right] \\ &= -\alpha s^2 - a_1 s \cdot \tanh \frac{s}{\varepsilon} - a_2 s \cdot sign\,(s)\,|s|^\tau \\ &= -\alpha s^2 - a_1 |s| \cdot \tanh \frac{|s|}{\varepsilon} - a_2 |s|^{\tau+1} \le 0 \end{aligned} \qquad (30)$$

The system is proven to achieve asymptotic convergence within finite time.

**Lower allocation design**

To achieve optimal torque distribution across all four wheels, this paper proposes the DWMEA method. The DWMEA method differs from traditional weighted minimum energy methods requiring iterative optimization. This approach provides a closed-form solution through KKT matrix inversion without iterative computation. Furthermore, its dynamic weight parameters can adaptively coordinate the effects of vertical loads, steering angles, and vehicle speeds. Aiming at minimizing weighted energy consumption, this method achieves efficient distribution of external yaw moments, ensuring that torque allocation to each wheel enhances handling stability while preventing wheel slip.

The energy consumption is characterized by the weighted sum of squared wheel torques. The objective function for minimizing weighted energy consumption is expressed as

$$\min J = \sum_{i=fl,fr,rl,rr} \omega_i T_i^2 \qquad (31)$$

The dynamic weight function is defined as

$$\omega_i = \eta_1 \frac{F_{z0}}{F_{i,z} + \varepsilon^*} + \eta_2 \frac{|\delta|}{\delta_0} \cdot I_{front}\,(i) + \eta_3 \frac{v_x}{v_0} \qquad (32)$$

where $\eta_1$ denotes the vertical load adjustment coefficient, $\eta_2$ represents the steering condition adjustment coefficient, $\eta_3$ is the velocity sensitivity adjustment coefficient, $F_{z0}$ is the nominal vertical load reference value, $\delta_0$ is the maximum steering angle reference value, $v_0$ is the nominal velocity reference value, $\varepsilon^*$ is the anti-singularity constant to prevent division by zero, and $I_{front}\,(i)$ serves as the differentiation index between front and rear axles for weight adjustment mechanisms. The specific expressions are given by:

$$I_{front}\,(i) = \begin{cases} 1, i = fl \text{ or } fr \\ 0, i = rl \text{ or } rr \end{cases} \qquad (33)$$

Considering the total longitudinal force and yaw moment constraints of the vehicle [32–35], this paper introduces the equality constraints:

$$\begin{cases} (T_{fl} + T_{fr}) \cos \delta + T_{rl} + T_{rr} = T_o \\ \frac{B_f}{2R} (T_{fr} - T_{fl}) \cos \delta + \frac{B_r}{2R} (T_{rr} - T_{rl}) = \Delta M_z \end{cases} \tag{34}$$

where $T_o$ denotes the total required longitudinal driving force of the vehicle.

Meanwhile, friction circle constraints for tire forces and actuator saturation constraints are added to ensure that the longitudinal and lateral forces on each tire do not exceed the maximum available friction, and the motor output torque is limited within a safe and reasonable range. Therefore, the following constraints are introduced:

$$\begin{cases} \sqrt{F_{i,x}^2 + F_{i,y}^2} \leq \mu F_{i,z} \\ |T_i| \leq T_{motor\_max} \end{cases}, (i = fl, fr, rl, rr) \tag{35}$$

Considering the aforementioned inequality constraints, penalty terms for friction circle constraints and actuator saturation constraints are incorporated into the dynamic weight function. The dynamic weight function is updated as follows:

$$\omega\prime_i = \omega_i \left(1 + \varsigma_1 \frac{\sqrt{F_{i,x}^2 + F_{i,y}^2}}{\mu F_{i,z}}\right) \left(1 + \varsigma_2 \frac{|F_{i,x}R|}{T_{motor\_max}}\right) \tag{36}$$

where $\varsigma_1$ and $\varsigma_2$ denote the constraint gains for friction circle constraints and actuator saturation constraints, respectively, controlling the penalty intensity when constraints approach upper limits, and $T_{motor\_max}$ represents the maximum motor torque.

A Lagrangian function incorporating dual constraints of total longitudinal force and yaw moment is constructed:

$$L = \sum_{i=fl,fr,rl,rr} \omega\prime_i T_i^2 + \lambda_1 [(T_{fl} + T_{fr}) \cos \delta + T_{rl} + T_{rr} - T_o] \\ + \lambda_2 [B_f (T_{fr} - T_{fl}) \cos \delta + B_r (T_{rr} - T_{rl}) - 2R\Delta M_z] \tag{37}$$

By taking partial derivatives of $L$ with respect to $T_{fl}$, $T_{fr}$, $T_{rl}$ and $T_{rr}$, and setting them to zero, we obtain:

$$\begin{cases} \frac{\partial L}{\partial T_{fl}} = 2\omega\prime_{fl} T_{fl} + \lambda_1 \cos \delta - \lambda_2 B_f \cos \delta = 0 \\ \frac{\partial L}{\partial T_{fr}} = 2\omega\prime_{fr} T_{fr} + \lambda_1 \cos \delta + \lambda_2 B_f \cos \delta = 0 \\ \frac{\partial L}{\partial T_{rl}} = 2\omega\prime_{rl} T_{rl} + \lambda_1 - \lambda_2 B_r = 0 \\ \frac{\partial L}{\partial T_{rr}} = 2\omega\prime_{rr} T_{rr} + \lambda_1 + \lambda_2 B_r = 0 \end{cases} \tag{38}$$

Combining Equations (34) and (36), the matrix equation $AX = B$ is formulated as

$$\begin{bmatrix} 2\omega\prime_{fl} & 0 & 0 & 0 & -B_f \cos \delta & \cos \delta \\ 0 & 2\omega\prime_{fr} & 0 & 0 & B_f \cos \delta & \cos \delta \\ 0 & 0 & 2\omega\prime_{rl} & 0 & -B_r & 1 \\ 0 & 0 & 0 & 2\omega\prime_{rr} & B_r & 1 \\ \cos \delta & \cos \delta & 1 & 1 & 0 & 0 \\ -B_f \cos \delta & B_f \cos \delta & -B_r & B_r & 0 & 0 \end{bmatrix} \begin{bmatrix} T_{fl} \\ T_{fr} \\ T_{rl} \\ T_{rr} \\ \lambda_2 \\ \lambda_1 \end{bmatrix} = \begin{bmatrix} 0 \\ 0 \\ 0 \\ 0 \\ T_0 \\ 2R\Delta M_z \end{bmatrix} \tag{39}$$

The KKT matrix $A$ can be expressed as

$$A = \begin{bmatrix} \Delta^2 J & \Delta g^T \\ \Delta g & 0 \end{bmatrix} \tag{40}$$

Since $\omega_i > 0 (i = fl, fr, rl, rr)$ (the Hessian matrix $\Delta^2 J$ is positive definite) and

$$\Delta g = \begin{bmatrix} \Delta g_1 \\ \Delta g_2 \end{bmatrix} = \begin{bmatrix} \cos\delta & \cos\delta & 1 & 1 \\ -B_f\cos\delta & B_f\cos\delta & -B_r & B_r \end{bmatrix} \tag{41}$$

Assume there exists a scalar $k^*$ satisfying $\Delta g_1 = k^*\Delta g_2$, which implies:

$$\begin{cases} \cos\delta = k^*(-B_f\cos\delta) \\ \cos\delta = k^*B_f\cos\delta \\ 1 = k^*(-B_r) \\ 1 = k^*B_r \end{cases} \tag{42}$$

For any front axle track $B_f > 0$, rear axle track $B_r > 0$, and front wheel steering angle $\delta \neq 90°$, Equation (42) yields $k^* = -1/B_f = 1/B_f = -1/B_r = 1/B_r$. This equation is clearly contradictory, indicating that no scalar $k^*$ satisfies the requirements. Therefore, regardless of vehicle parameter variations, provided the physical constraints ($B_f > 0$, $B_r > 0$, $\delta \neq 90°$) are satisfied, the vectors $\Delta g_1$ and $\Delta g_2$ are linearly independent, and the matrix $\Delta g$ is row full-rank. Combining this with the previously proven positive definiteness of the Hessian matrix $\Delta^2 J$, the KKT matrix $A$ must be nonsingular. Consequently, the solution to the matrix equation (39) is unique.

Incorporating Equation (39), the unique solution containing the four-wheel torques is derived as

$$\begin{bmatrix} T_{fl} \\ T_{fr} \\ T_{rl} \\ T_{rr} \\ \lambda_2 \\ \lambda_1 \end{bmatrix} = \begin{bmatrix} 2\omega_{fl} & 0 & 0 & 0 & -B_f\cos\delta & \cos\delta \\ 0 & 2\omega_{fr} & 0 & 0 & B_f\cos\delta & \cos\delta \\ 0 & 0 & 2\omega_{rl} & 0 & -B_r & 1 \\ 0 & 0 & 0 & 2\omega_{rr} & B_r & 1 \\ \cos\delta & \cos\delta & 1 & 1 & 0 & 0 \\ -B_f\cos\delta & B_f\cos\delta & -B_r & B_r & 0 & 0 \end{bmatrix}^{-1} \begin{bmatrix} 0 \\ 0 \\ 0 \\ 0 \\ T_0 \\ 2R\Delta M_z \end{bmatrix} \tag{43}$$

## Simulation results and analyses

The DYC control system developed in this study adopts a hierarchical architecture: the upper-layer controller employs the AEWC-SMC to obtain the required external yaw moment, while the lower-layer torque allocation module distributes four-wheel torques based on the DWMEA method. For brevity in subsequent descriptions, this DYC control architecture will be referred to as the AEWC-SMC scheme. To evaluate the effectiveness of the control system, a simulation platform was constructed based on the MATLAB/Simulink environment. Sine Condition and Fishhook Condition were selected for validation and compared with NTSMC (Nonsingular Terminal Sliding Mode Control) [23], SMC, MPC (Model Predictive Control), Fuzzy PID control, and no-control scheme, where vehicle fundamental parameters are consistent with those in this paper's scheme. To more closely approximate real-world driving scenarios, two critical enhancements were implemented: 1. Injection of Gaussian white noise into the controller state feedback channel, adhering to automotive-grade IMU accuracy specifications (±0.5° yaw rate/sideslip angle, ±0.2m/s longitudinal velocity). 2. Dynamic enforcement of tire force

constraints via stepwise calculation of $\mu_{available} = \sqrt{1 - \left(\frac{F_y}{\mu F_z}\right)^2}$, ensuring longitudinal demanded forces do not exceed physical boundaries of the friction circle. To account for actuator dynamics, a first-order delay model with time constant $\tau\prime = 100ms$ was applied to the yaw moment output of the upper-layer controller, reflecting cumulative delays in motion control systems. For wheel torque actuation, a shorter time constant of 50ms was adopted based on modern hub motor dynamics. The compensated output is:

$$T_{actual}(k) = \alpha\prime T_{actual}(k-1) + (1 - \alpha\prime) T_{cmd}(k) \tag{44}$$

where $\alpha\prime = e^{-\Delta t/\tau\prime}$ denotes the delay coefficient, $\Delta t$ the sampling period, $\tau\prime$ the motor time constant, $T_{actual}$ the actual output torque, and $T_{cmd}$ the commanded torque.

Regarding parameter selection in the controller, this study conducted a parameter sensitivity analysis on key parameters $\lambda$, $\kappa$, $\alpha$, $a_1$, $a_2$, $\eta_1$, $\eta_2$ and $\eta_3$. The sideslip angle Mean Absolute Error (MAE) and yaw rate Mean Absolute Error were utilized as sensitivity indicators. For each parameter subjected to analysis, corresponding scanning intervals were defined individually while fixing other parameters. Simulation was executed per cycle, performance metrics were recorded, and impacts on sideslip angle MAE and yaw rate MAE were comprehensively evaluated. This resulted in a single-variable sensitivity analysis table, as presented in Table 1.

The MATLAB/Simulink vehicle simulation parameters in this study were calibrated based on data from an actual vehicle test platform. The experimental platform utilizes a four-wheel independently driven electric vehicle prototype, with mass parameters dynamically calibrated using a vehicle center-of-gravity tester, and tire stiffness parameters fitted according to measured data from an MTS Flat-Trac test bench. Based on the foregoing, the fundamental vehicle parameters, AEWC-SMC controller parameters, and DWMEA allocator parameters are detailed in Table 2. The steering wheel angle ($\theta_{sw}$) input signals under Sine Condition and Fishhook Condition operating conditions are shown in Figs 4 and 5, respectively.

### Sine condition

Under the Sine Condition, two test cases with different vehicle longitudinal speeds and road adhesion coefficients were employed: "Case 1" and"Case 2". " Case 1 " test case set the vehicle longitudinal velocity to $22m/s$ and the road adhesion coefficient to 0.3. "Case 2" test case set the vehicle longitudinal velocity to $33m/s$ and the road adhesion coefficient to 0.8.

Fig 6 displays the simulation results under the " Case 1" test scenario. From the comparison of sideslip angle and sideslip angle tracking errors in Fig 6(a) and 6(b), it can be observed that the AEWC-SMC scheme exhibits superior sideslip angle tracking accuracy compared to other control schemes, while the no-control scheme fails to ensure tracking

**Table 1. Single-variable sensitivity analysis.**

| Parameter | Optimal Value | Sideslip Angle MAE | Yaw Rate MAE |
|---|---|---|---|
| $\lambda$ | 0.02 | 0.471 | 0.131 |
| $\kappa$ | 53 | 0.457 | 0.119 |
| $\alpha$ | 14 | 0.417 | 0.116 |
| $a_1$ | 8 | 0.412 | 0.116 |
| $a_2$ | 5 | 0.409 | 0.113 |
| $\eta_1$ | 1.1 | 0.382 | 0.105 |
| $\eta_2$ | 0.7 | 0.377 | 0.099 |
| $\eta_3$ | 0.3 | 0.372 | 0.095 |

**Table 2. Simulation parameters.**

| Vehicle Parameters | Value | AEWC-SMC Control Parameters | Value | DWMEA Parameters | Value |
|---|---|---|---|---|---|
| $m$ | 1765 $kg$ | $\lambda$ | 0.02 | $\eta_1$ | 1.1 |
| $m_w$ | 41.25 $kg$ | $\kappa$ | 53 | $\eta_2$ | 0.7 |
| $I_z$ | 2700 $kg \cdot m^2$ | $\alpha$ | 14 | $\eta_3$ | 0.3 |
| $h_g$ | 0.5 $m$ | $a_1$ | 8 | $\varepsilon^*$ | $1e-6$ |
| $B = B_f = B_r$ | 1.6 $m$ | $a_2$ | 5 | $F_{z0}$ | 4324.25 $N$ |
| $L_f$ | 1.2 $m$ | $\varepsilon$ | 0.08 | $\delta_0$ | $\frac{40*\pi}{180}$ $rad$ |
| $L_r$ | 1.4 $m$ | $\tau$ | $\begin{cases} 0.55, \theta_{sw} = 0 \\ 0.25, \left\|\theta_{sw}\right\| > 0 \end{cases}$ | $v_0$ | 22 $m/s$ |
| $k_f$ | $-200e3$ $N/rad$ | | | $\varsigma_1$ | 0.5 |
| $k_r$ | $-200e3$ $N/rad$ | | | $\varsigma_2$ | 0.5 |
| $T_{motor\_max}$ | 1000 $N \cdot m$ | | | | |

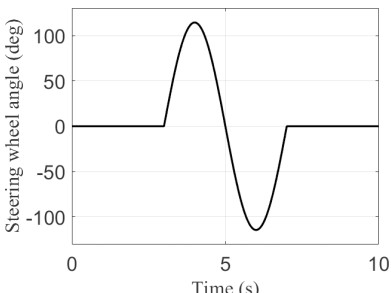

**Fig 4. Sine condition.**

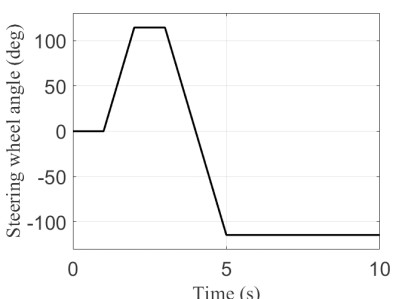

**Fig 5. Fishhook condition.**

precision of the sideslip angle. From the yaw rate and yaw rate tracking errors in Fig 6(c) and 6(d), it is evident that the yaw rate tracking accuracy of the AEWC-SMC scheme significantly outperforms that of other control schemes. The phase plane plot in Fig 6(e) reveals that the AEWC-SMC scheme has a smaller convergence range and higher stability margin than other control schemes, and the no-control scheme cannot guarantee vehicle handling stability. Concurrently, Table 3 comprehensively presents the MAE (Mean Absolute Error), RMSE (Root Mean Square Error), and Standard Deviation of the yaw rate, demonstrating that the AEWC-SMC scheme provides enhanced lateral stability. Fig 6(f) and 6(g) illustrate

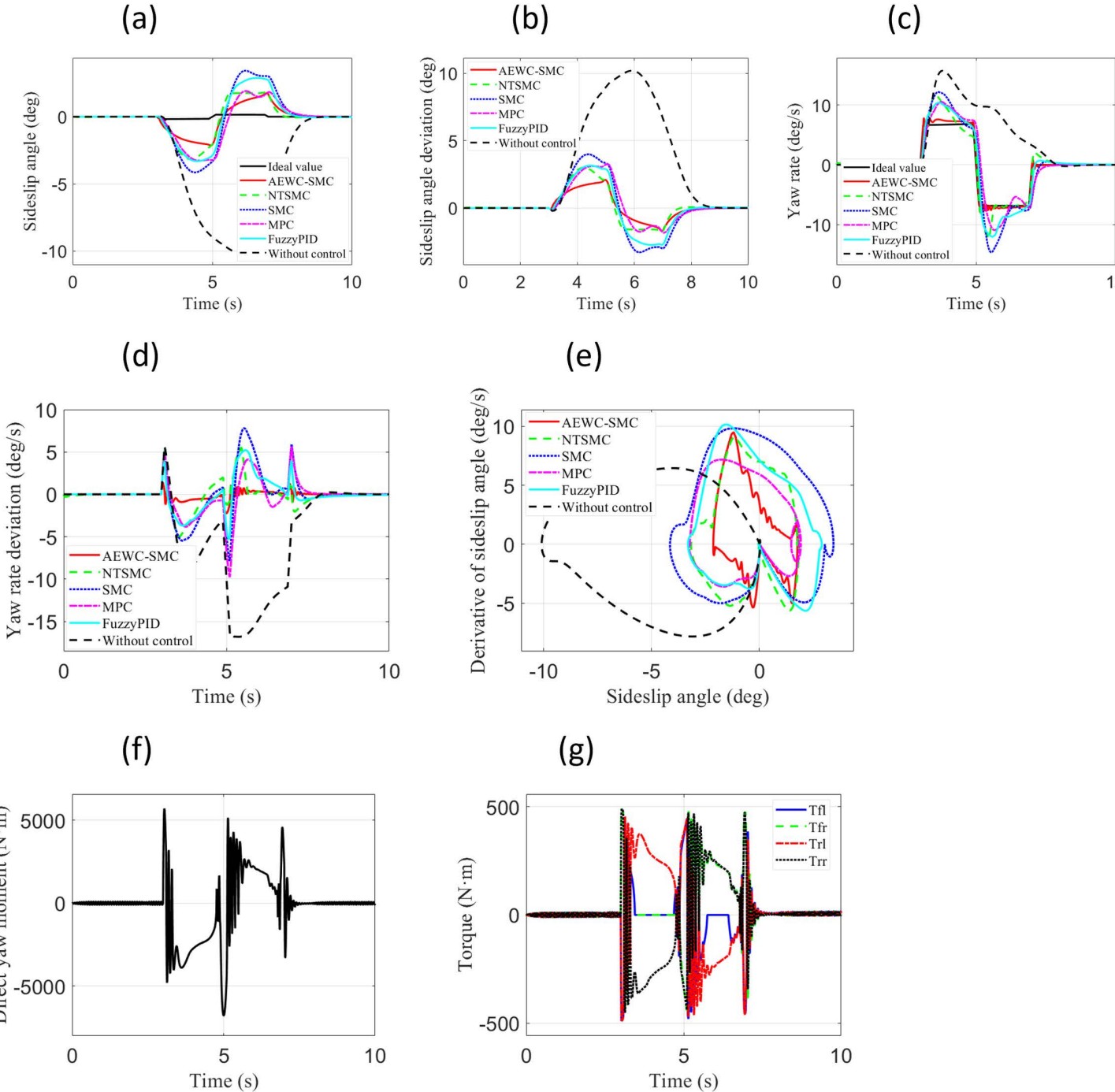

**Fig 6. Comparative simulation results for "Case 1".** (a) Sideslip angle; (b) Sideslip angle error; (c) Yaw rate; (d) Yaw rate error; (e) $\beta - \dot{\beta}$ Phase plane; (f) External yaw moment; (g) Four-wheel torque.

**Table 3. Yaw rate tracking errors under sine condition.**

| Strategy | MAE | RMSE | Standard Deviation |
|---|---|---|---|
| AEWC_SMC ($\mu = 0.3$) | 0.228207201 | 0.430039897 | 0.426178555 |
| NTSMC ($\mu = 0.3$) | 0.759758438 | 1.511630957 | 1.508938047 |
| SMC ($\mu = 0.3$) | 1.302263346 | 2.470967502 | 2.471441773 |
| MPC ($\mu = 0.3$) | 1.049054283 | 1.976214998 | 1.96692804 |
| Fuzzy PID ($\mu = 0.3$) | 0.988156386 | 1.720967412 | 1.721825021 |
| AEWC_SMC ($\mu = 0.8$) | 0.529351048 | 1.275722689 | 1.237690046 |
| NTSMC ($\mu = 0.8$) | 3.235163949 | 5.580298202 | 5.578672713 |
| SMC ($\mu = 0.8$) | 1.794071258 | 3.367356435 | 3.367692238 |
| MPC ($\mu = 0.8$) | 1.751117563 | 3.310922502 | 3.312294218 |
| Fuzzy PID ($\mu = 0.8$) | 1.384702488 | 2.539438729 | 2.540392793 |

the external yaw moment output from the upper-layer controller and the four-wheel torque output from the lower-layer allocation in the proposed AEWC-SMC scheme, demonstrating the validity of the control outputs.

Fig 7 shows the simulation results for the " Case 2" test scenario. From Fig 7(a)–7(d), it is evident that the AEWC-SMC scheme demonstrates significantly superior tracking accuracy for both sideslip angle and yaw rate compared to other control schemes. Fig 7(e) indicates that the AEWC-SMC scheme exhibits significant advantages in terms of smaller convergence range and higher stability margin, while Table 3 demonstrates better lateral stability of the AEWC-SMC scheme based on MAE, RMSE, and Standard Deviation of the yaw rate. Fig 7(f) and 7(g) display the external yaw moment output from the upper-layer controller and the four-wheel torque output from the lower-layer allocation in the proposed AEWC-SMC scheme, validating the effectiveness of the control outputs.

Simulation results of both test cases under the Sine Condition demonstrate that the proposed AEWC-SMC scheme achieves higher sideslip angle tracking precision, higher yaw rate tracking precision, smaller convergence ranges, and higher stability margins overall. These collectively exhibit significant superiority in comprehensive control performance.

### Fishhook condition

Under the Fishhook Condition, two test cases with different vehicle longitudinal velocities and road adhesion coefficients were employed: "Case 3" and"Case 4". " Case 3 " test case set the vehicle longitudinal velocity to $22m/s$ and the road adhesion coefficient to 0.3. " Case 4 " test case set the vehicle longitudinal velocity to $33m/s$ and the road adhesion coefficient to 0.8.

Fig 8 presents the simulation results for the " Case 3" test scenario. From the comparison of sideslip angle and sideslip angle tracking errors in Fig 8(a) and 8(b), it can be observed that the AEWC-SMC scheme exhibits superior sideslip angle tracking accuracy compared to other control schemes, while the no-control scheme fails to ensure tracking precision of the sideslip angle. From the yaw rate and yaw rate tracking errors in Fig 8(c) and 8(d), it is evident that the yaw rate tracking accuracy of the AEWC-SMC scheme significantly outperforms that of other control schemes. The phase plane plot in Fig 8(e) reveals that the AEWC-SMC scheme has a smaller convergence range and higher stability margin than other control schemes, and the no-control scheme cannot guarantee vehicle handling stability. Concurrently, Table 4 comprehensively presents the MAE, RMSE, and Standard Deviation of the yaw rate, demonstrating the enhanced lateral stability of the AEWC-SMC scheme. Fig 8(f) and 8(g) show the external yaw moment output from the upper-layer controller and the four-wheel torque output from the lower-layer allocation in the proposed AEWC-SMC scheme, demonstrating the validity of the control outputs.

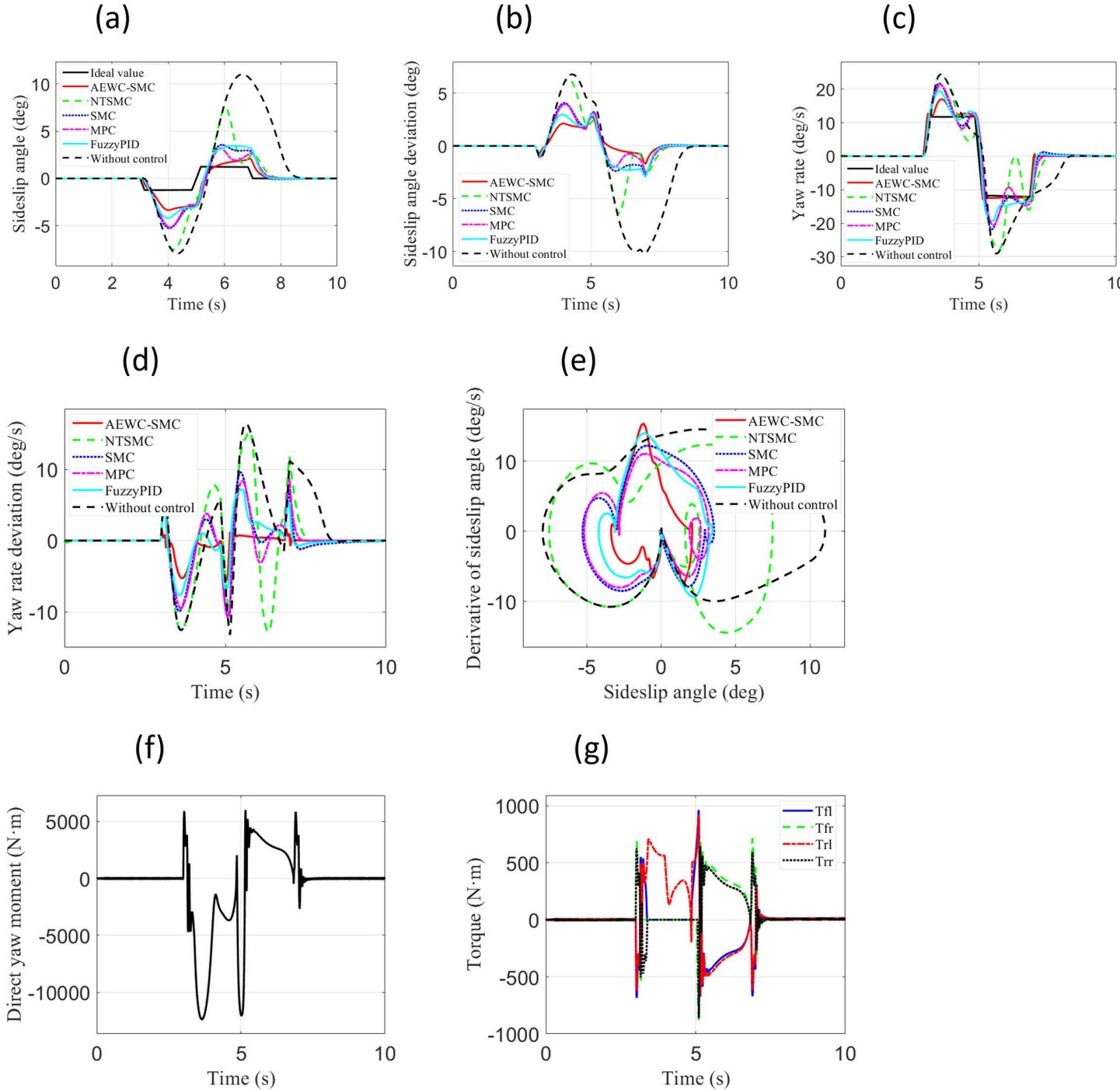

**Fig 7. Comparative simulation results for "Case 2".** (a) Sideslip angle; (b) Sideslip angle error; (c) Yaw rate; (d) Yaw rate error; (e) $\beta - \dot{\beta}$ Phase plane; (f) External yaw moment; (g) Four-wheel.

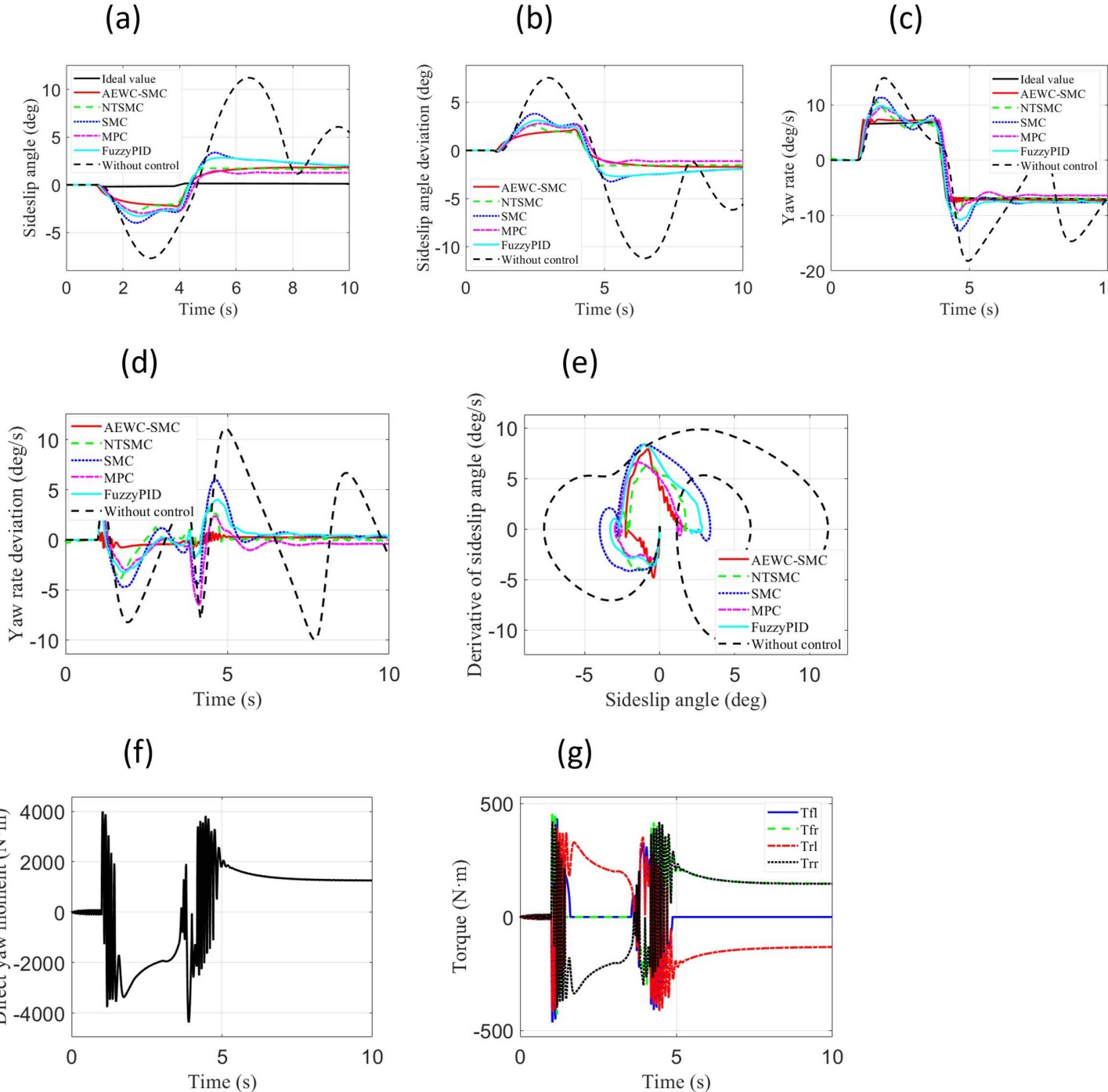

**Fig 8. Comparative simulation results for "Case 3".** (a) Sideslip angle; (b) Sideslip angle error; (c) Yaw rate; (d) Yaw rate error; (e) $\beta - \dot{\beta}$ Phase plane; (f) External yaw moment; (g) Four-wheel.

**Table 4. Yaw rate tracking errors under fishhook condition.**

| Strategy | MAE | RMSE | Standard Deviation |
|---|---|---|---|
| AEWC_SMC ($\mu = 0.3$) | 0.305364339 | 0.361635822 | 0.361410195 |
| NTSMC ($\mu = 0.3$) | 0.570895556 | 1.128974371 | 1.129329408 |
| SMC ($\mu = 0.3$) | 1.367412163 | 2.066737596 | 2.056455443 |
| MPC ($\mu = 0.3$) | 0.958430827 | 1.477278046 | 1.347120844 |
| Fuzzy PID ($\mu = 0.3$) | 1.003602757 | 1.409292049 | 1.383543717 |
| AEWC_SMC ($\mu = 0.8$) | 0.609342754 | 0.987032218 | 0.967256238 |
| NTSMC ($\mu = 0.8$) | 6.990674384 | 9.260079705 | 9.262505313 |
| SMC ($\mu = 0.8$) | 2.590777996 | 3.409297134 | 3.384211029 |
| MPC ($\mu = 0.8$) | 2.232868583 | 3.218084044 | 3.211241637 |
| Fuzzy PID ($\mu = 0.8$) | 1.955207126 | 2.456388656 | 2.382851956 |

Fig 9 shows the simulation results for the " Case 4" test scenario. From Fig 9(a)–9(d), it is evident that the AEWC-SMC scheme demonstrates significantly superior tracking accuracy for both sideslip angle and yaw rate compared to other control schemes. Fig 8(e) indicates that the AEWC-SMC scheme exhibits significant advantages in terms of smaller convergence range and higher stability margin, while Table 4 demonstrates better lateral stability of the AEWC-SMC scheme based on MAE, RMSE, and Standard Deviation of the yaw rate. Fig 8(f) and 8(g) display the external yaw moment output from the upper-layer controller and the four-wheel torque output from the lower-layer allocation in the proposed AEWC-SMC scheme, validating the effectiveness of the control outputs.

Simulation results for both test cases under the Fishhook Condition similarly demonstrate that the proposed AEWC-SMC scheme exhibits higher sideslip angle tracking accuracy, higher yaw rate tracking accuracy, smaller convergence ranges, and higher stability margins overall. These collectively manifest significant superiority in comprehensive control performance.

## Conclusions

This study designs a novel DYC scheme. For the upper-layer control strategy, an AEWC-SMC is designed. By introducing a nonlinear weighting factor into the sliding surface, rapid response to large sideslip angle deviations and linear control for minor deviations are achieved, enabling adaptive precision control. Simultaneously, in the reaching law design, linear terms, smooth nonlinear terms, and fractional-order nonlinear terms are incorporated to construct a composite reaching law, allowing the controller to achieve rapid correction under large errors, smooth transition under small errors, and thereby fast convergence with reduced chattering. For the lower-layer allocation design, a DWMEA method requiring no iterative computation is proposed. By formulating an objective function to minimize weighted energy consumption, considering the effects of vertical load, steering angle, and vehicle speed, and introducing adaptive dynamic weight parameters, optimal four-wheel torque distribution is adaptively realized. The simulations conducted under both Sine Condition and Fishhook Condition across four test cases demonstrate that the proposed control scheme achieves higher tracking accuracy, faster convergence speed, and improved stability, possessing substantial practical value. Future work will conduct hardware-in-the-loop (HIL) validation and real-vehicle verification, covering more complex fault conditions, with experimental evaluation of the proposed control strategy focused on physical vehicles.

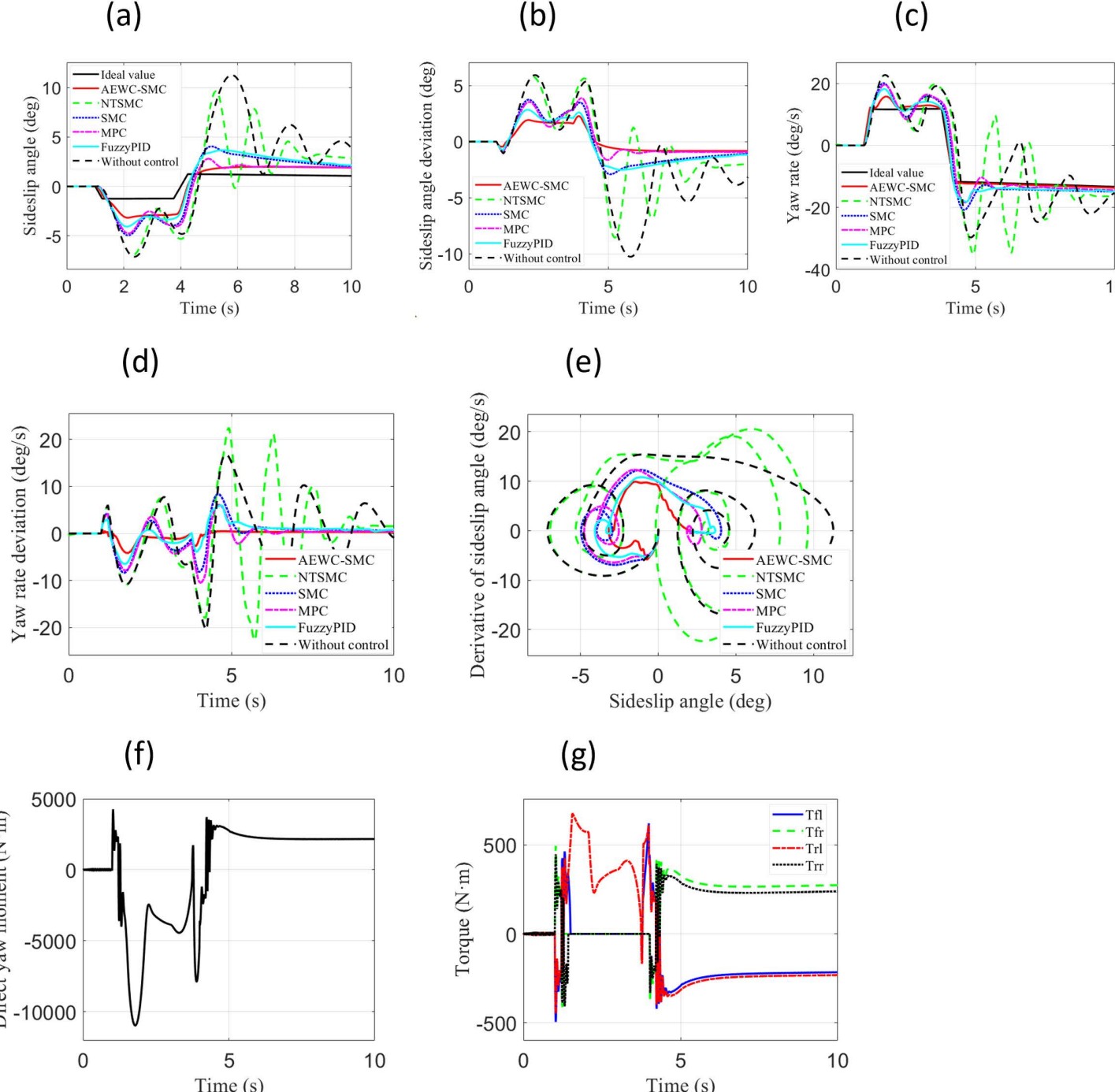

**Fig 9. Comparative simulation results for "Case 4".** (a) Sideslip angle; (b) Sideslip angle error; (c) Yaw rate; (d) Yaw rate error; (e) $\beta - \dot{\beta}$ Phase plane; (f) External yaw moment; (g) Four-wheel.

## Author contributions

**Conceptualization:** Zhengyong Tao, Mingming Wu, Zhongzhi Tong.

**Formal analysis:** Banglai Sun.

**Methodology:** Zhengyong Tao, Mingming Wu, Deqiang Xie, Zhongzhi Tong.

**Resources:** Min Qu, Hui Wu, Banglai Sun.

**Software:** Zhengyong Tao.

**Validation:** Zhengyong Tao, Min Qu, Hui Wu, Banglai Sun, Deqiang Xie.

**Writing – original draft:** Zhengyong Tao.

**Writing – review & editing:** Zhengyong Tao, Zhongzhi Tong.

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
