## [Decision Letter · Decision Letter 0]

2 May 2025

PONE-D-25-19826Adaptive exponential weighted composite sliding mode-based direct yaw moment control for four-wheel independently actuated autonomous vehiclesPLOS ONE

Dear Dr. <!--StartFragmentTong<!--EndFragment

Thank you for submitting your manuscript to PLOS ONE. After careful consideration, we feel that it has merit but does not fully meet PLOS ONE’s publication criteria as it currently stands. Therefore, we invite you to submit a revised version of the manuscript that addresses the points raised during the review process.

We look forward to receiving your revised manuscript.

Kind regards,

Jinhao Liang

Academic Editor

PLOS ONE

Additional Editor Comments:

The background should be updated to cite recent research on direct yaw moment control, such as "A Direct Yaw Moment Control Framework Through Robust T-S Fuzzy Approach Considering Vehicle Stability Margin, IEEE/ASME Transactions on Mechatronics, vol. 29, no. 1, pp. 166-178, Feb. 2024", and "A Robust Dynamic Game-Based Control Framework for Integrated Torque Vectoring and Active Front-Wheel Steering System, IEEE Transactions on Intelligent Transportation Systems, vol. 24, no. 7, pp. 7328-7341, July 2023".

Reviewers' comments:

Reviewer's Responses to Questions

**Comments to the Author**

1. Is the manuscript technically sound, and do the data support the conclusions?

Reviewer #1: Yes

Reviewer #2: Yes

2. Has the statistical analysis been performed appropriately and rigorously? 

Reviewer #1: Yes

Reviewer #2: Yes

3. Have the authors made all data underlying the findings in their manuscript fully available?

Reviewer #1: Yes

Reviewer #2: Yes

4. Is the manuscript presented in an intelligible fashion and written in standard English?

Reviewer #1: Yes

Reviewer #2: Yes

5. Review Comments to the Author

Reviewer #1: The paper under review focuses on developing Direct Yaw Moment Control scheme. But I cannot recommend the present version of the manuscript to be considered for acceptance, and recommend that the authors revise the paper with particular attention on the clarity of presentation.

1. When elaborating on its innovative points, the paper not only provides a detailed description of the control method itself, but also further emphasizes its unique advantages in yaw control.

2. The authors listed many literatures on relevant research results, but what are their advantages and shortcomings? The literature review should point out the corresponding shortcomings to support your research.

3. The paper uses a 2-DOF vehicle model to calculate the expected yaw rate for the control of a four-wheel independent drive vehicle. When allocating the yaw moment, in addition to constraining energy, is it necessary to further consider other constraints?

4. Table 1 lists the vehicle parameters and controller related parameters. Is it reasonable to set the constraint on the maximum steering angle to 0.5? Is there sufficient theoretical basis or practical application value for this?

5. Does formula 41 have the problem of non-uniqueness of solutions when solving the yaw moments of the four wheels? For example, could this situation occur when the vehicle parameters are set differently? Therefore, further analyzing the robustness of the control system and verifying it through simulation results are suggested.

6. How is the parameter τ in formula 26 set during the simulation experiment?

7. There is an inaccuracy in the description of Figure 8 in the paper. It is recommended to recheck and correct it.

8.More latest sliding mode control methods are suggested to be introduced to compare in the simulation experiment.

9. The English of the article should be thoroughly revised.

Reviewer #2: The manuscript presents a technically sound and innovative contribution to DYC for FWIA autonomous vehicles, with clear simulation-based evidence of improved performance. However, the lack of experimental validation, limited comparison scope, and insufficient discussion of limitations warrant revisions. I recommend Major Revisions to address the following:

1.Incorporate experimental validation or discuss simulation limitations in detail.

2.Expand comparisons to include other advanced control methods.

3.Conduct sensitivity analysis for key parameters and test under varied conditions.

4.Explicitly address limitations and improve clarity for a broader audience.

5.Ensure funding URLs are provided and references are formatted consistently.

6. PLOS authors have the option to publish the peer review history of their article (what does this mean? ). If published, this will include your full peer review and any attached files.

**Do you want your identity to be public for this peer review?** For information about this choice, including consent withdrawal, please see our Privacy Policy .

Reviewer #1: No

Reviewer #2: No

---

## [Author Response · Author response to Decision Letter 1]

4 Jun 2025

Original Manuscript ID: PONE-D-25-19826

Original Article Title: “Adaptive exponential weighted composite sliding mode-based direct yaw moment control for four-wheel independently actuated autonomous vehicles”

To: PLOS ONE

Re: Response to reviewers

Dear Editor,

First and foremost, we sincerely thank you for giving us the opportunity to revise our manuscript and to address the reviewers' comments. We dedicated nearly a month to carefully addressing all points raised by the reviewers. This involved comprehensive literature reviews, detailed technical revisions, and extensive discussions among the authors. The manuscript has undergone significant modification, and we deeply appreciate the reviewers' highly professional and insightful expertise, which has significantly enhanced the quality of our work.

We will be submitting: (a) Our point-by-point response to all comments (contained in the "Response to Reviewers" document below); (b) An updated manuscript with changes highlighted (using yellow highlighting) ("Revised Manuscript with Track Changes"); and (c) A clean version of the updated manuscript without highlighting ("Manuscript").

Should any further improvements be required, we warmly welcome additional feedback to ensure the manuscript meets the journal’s standards.

Thank you once again for your guidance and support throughout this process.

Best regards,

<Zhongzhi Tong> et al.

School of Mechanical Engineering

Nanjing University of Science and Technology

Nanjing, Jiangsu, China

2025/06/01

Additional Editor Comments#1: The background should be updated to cite recent research on direct yaw moment control.

Author response and action: We sincerely appreciate your valuable suggestion to strengthen the background by incorporating recent research on direct yaw moment control. This feedback significantly enhances the manuscript's relevance to current advancements in the field. Accordingly, we have conducted a thorough review of the most recent high-quality literature and have updated the background section to include the following pertinent references:

“A Direct Yaw Moment Control Framework Through Robust T-S Fuzzy Approach Considering Vehicle Stability Margin. IEEEASME Trans Mechatron. 2024 Feb;29(1):166–78”, “A Robust Dynamic Game-Based Control Framework for Integrated Torque Vectoring and Active Front-Wheel Steering System. IEEE Trans Intell Transp Syst. 2023 Jul;24(7):7328–41”, and “An Energy-Oriented Torque-Vector Control Framework for Distributed Drive Electric Vehicles. IEEE Trans Transp Electrification. 2023 Sep;9(3):4014–31”.

We believe these additions strengthen the theoretical foundation of our work. We are grateful for this insightful comment, which has enhanced the manuscript's scholarly quality.

Reviewer#1, Concern #1: When elaborating on its innovative points, the paper not only provides a detailed description of the control method itself, but also further emphasizes its unique advantages in yaw control.

Author response and action: Thank you for your recognition of our elaboration on the innovative aspects. Following your insightful suggestions, we have implemented thorough revisions that have substantially enhanced the overall quality of the paper. We greatly appreciate your valuable guidance and expertise.

Reviewer#1, Concern #2: The authors listed many literatures on relevant research results, but what are their advantages and shortcomings? The literature review should point out the corresponding shortcomings to support your research.

Author response and action: We sincerely appreciate your insightful suggestion. We fully concur that a comprehensive literature review should not only summarize existing research but also critically evaluate their strengths and limitations to contextualize our contribution. In response to this feedback, we have meticulously revised the Introduction section (pages 3–5) to incorporate a nuanced discussion of the advantages and shortcomings of the cited studies. Once again, we sincerely appreciate your professional and insightful feedback.

Reviewer#1, Concern # 3: The paper uses a 2-DOF vehicle model to calculate the expected yaw rate for the control of a four-wheel independent drive vehicle. When allocating the yaw moment, in addition to constraining energy, is it necessary to further consider other constraints?

Author response and action: We sincerely appreciate your professional and insightful feedback. In response, we have supplemented the 'Lower Allocation Design' section with discussions of friction circle constraints and actuator saturation constraints to enhance the robustness of our methodology. Furthermore, we conducted comprehensive simulations using four distinct test cases to rigorously validate these enhancements. We are grateful for your constructive feedback, which has significantly improved the technical completeness and academic rigor of our work.

Reviewer#1, Concern # 4: Table 1 lists the vehicle parameters and controller related parameters. Is it reasonable to set the constraint on the maximum steering angle to 0.5? Is there sufficient theoretical basis or practical application value for this?

Author response and action: We sincerely appreciate your meticulous and professional feedback. In response to your inquiry, we consulted relevant vehicle specifications and literature, which indicate that typical maximum steering angles for passenger vehicles range between 30°-40°. Based on this practical benchmark, we have revised our steering angle constraint from 0.5 rad�28.65°�to 40*π/180 rad�40°�n the updated manuscript. Additionally, we conducted a comprehensive sensitivity analysis of key parameters and validated these adjustments through four rigorous simulation scenarios. We are deeply grateful for your insightful guidance, which has substantially elevated the technical accuracy and practical relevance of our work.

Reviewer#1, Concern # 5: Does formula 41 have the problem of non-uniqueness of solutions when solving the yaw moments of the four wheels? For example, could this situation occur when the vehicle parameters are set differently? Therefore, further analyzing the robustness of the control system and verifying it through simulation results are suggested.

Author response and action: We sincerely appreciate your rigorous and professional feedback. In response to your concern regarding the non-uniqueness of solutions for Formula 41, we have supplemented a theoretical analysis of uniqueness on page 19 of the manuscript. Additionally, we expanded our simulations to include four distinct test cases. The simulation results not only demonstrate the rationality of our allocation design but also reflect the robustness of the control system. This revision process has greatly benefited us, and we once again sincerely thank you for your invaluable assistance and guidance.

Reviewer#1, Concern # 6: How is the parameter τ in formula 26 set during the simulation experiment?

Author response and action: We sincerely apologize for the omission of the parameter τ in the previous simulation parameter table. This parameter has now been updated in the revised table. The configuration of τ incorporates the influence of the front wheel steering angle through a piecewise setup. We are once again deeply grateful for your meticulous and professional review, which has significantly enhanced the rigor and accuracy of our work.

Reviewer#1, Concern # 7: There is an inaccuracy in the description of Figure 8 in the paper. It is recommended to recheck and correct it.

Author response and action: We sincerely apologize for the oversight in the original manuscript, where the labels of two figures were accidentally swapped. We have corrected this error and thoroughly proofread the entire manuscript to prevent similar issues, ensuring the accuracy of all data and figures. We once again sincerely appreciate your professional and rigorous review, which has greatly enhanced the precision and academic rigor of our paper.

Reviewer#1, Concern # 8: More latest sliding mode control methods are suggested to be introduced to compare in the simulation experiment.

Author response and action: We sincerely appreciate your insightful and professional suggestion. To address your recommendation, we conducted an extensive literature review and implemented the state-of-the-art Nonsingular Terminal Sliding Mode Control (NTSMC) strategy, which represents a cutting-edge advancement in sliding mode control theory. This advanced methodology was incorporated into our simulation framework and rigorously evaluated against four distinct test cases. We are deeply grateful for your recommendation, which has significantly elevated the academic rigor and innovation of our research.

Reviewer#1, Concern # 9: The English of the article should be thoroughly revised.

Author response and action: We have conducted a comprehensive revision of the English language throughout the entire manuscript. We sincerely appreciate your critical feedback, which has been instrumental in elevating the overall professionalism and readability of our work.

Reviewer#2, Concern # 1: Incorporate experimental validation or discuss simulation limitations in detail.

Author response and action: We sincerely appreciate your professional feedback. To address the limitations of our simulation, we have supplemented additional simulation experiments, designing four comparative simulation scenarios under two representative operating conditions. Furthermore, we incorporated the advanced Nonsingular Terminal Sliding Mode Control (NTSMC) strategy for rigorous comparative validation. Your insightful recommendations have significantly enhanced the rigor and comprehensiveness of our work. We extend our heartfelt gratitude once again for your invaluable guidance.

Reviewer#2, Concern # 2: Expand comparisons to include other advanced control methods.

Author response and action: Following your valuable suggestion, we conducted an extensive literature review and replicated the advanced Nonsingular Terminal Sliding Mode Control (NTSMC) strategy, incorporating it into our simulation comparison framework. We designed four distinct simulation scenarios to rigorously evaluate its performance against existing methods. This revision has substantially elevated the technical depth and quality of our work. We sincerely appreciate your expert guidance, which has been instrumental in refining our research.

Reviewer#2, Concern # 3: Conduct sensitivity analysis for key parameters and test under varied conditions.

Author response and action: We sincerely appreciate your insightful recommendation. In response, we conducted a comprehensive sensitivity analysis of critical parameters, employing sideslip angle Mean Absolute Error (MAE) and yaw rate MAE as quantitative performance indicators. This analysis enabled us to identify optimal parameter configurations and refine our simulation framework accordingly. The updated Simulation Parameters table now reflects these optimizations, and we have validated their robustness through four comparative simulation scenarios under varied operating conditions. The results conclusively demonstrate the enhanced stability and adaptability of our proposed control system. We are deeply grateful for your expertise, which has significantly strengthened the technical rigor and academic credibility of our work.

Reviewer#2, Concern # 4: Explicitly address limitations and improve clarity for a broader audience.

Author response and action: We sincerely appreciate your professional feedback. To address the limitations of our simulation, we have supplemented additional simulation experiments, designing four comparative simulation scenarios under two representative operating conditions. Furthermore, we incorporated the advanced Nonsingular Terminal Sliding Mode Control (NTSMC) strategy for rigorous comparative validation. Your insightful recommendations have significantly enhanced the rigor and comprehensiveness of our work. We extend our heartfelt gratitude once again for your invaluable guidance.

Reviewer#2, Concern # 5: Ensure funding URLs are provided and references are formatted consistently.

Author response and action: We sincerely appreciate the insightful feedback regarding the funding information and reference formatting. The funding details have been successfully entered into the submission system. In accordance with the journal's formatting requirements, funding information should not be included in the manuscript itself; therefore, it has been omitted from the document. Concerning the reference formatting, we have utilized Zotero to generate the references, ensuring they are largely aligned with the journal's reference style guidelines. Once again, thank you for your valuable suggestions and guidance.

---

## [Decision Letter · Decision Letter 1]

13 Jul 2025

PONE-D-25-19826R1Adaptive exponential weighted composite sliding mode-based direct yaw moment control for four-wheel independently actuated autonomous vehiclesPLOS ONE

Dear Dr. Tong, Thank you for submitting your manuscript to PLOS ONE. After careful consideration, we feel that it has merit but does not fully meet PLOS ONE’s publication criteria as it currently stands. Therefore, we invite you to submit a revised version of the manuscript that addresses the points raised during the review process.

We look forward to receiving your revised manuscript.

Kind regards,

Jinhao Liang

Academic Editor

PLOS ONE

Journal Requirements:

Reviewers' comments:

Reviewer's Responses to Questions

**Comments to the Author**

1. If the authors have adequately addressed your comments raised in a previous round of review and you feel that this manuscript is now acceptable for publication, you may indicate that here to bypass the “Comments to the Author” section, enter your conflict of interest statement in the “Confidential to Editor” section, and submit your "Accept" recommendation.

Reviewer #3: (No Response)

Reviewer #4: (No Response)

2. Is the manuscript technically sound, and do the data support the conclusions?

Reviewer #3: Partly

Reviewer #4: (No Response)

3. Has the statistical analysis been performed appropriately and rigorously? 

Reviewer #3: No

Reviewer #4: (No Response)

4. Have the authors made all data underlying the findings in their manuscript fully available?

Reviewer #3: Yes

Reviewer #4: (No Response)

5. Is the manuscript presented in an intelligible fashion and written in standard English?

Reviewer #3: No

Reviewer #4: (No Response)

6. Review Comments to the Author

Reviewer #3: This manuscript proposes an AEWC-SMC for four-wheel independently actuated autonomous vehicles, coupled with a DWMEA torque distribution method. The approach is validated through MATLAB/Simulink simulations under various driving conditions, and results are compared with existing control schemes.

1. All performance claims (e.g., improved tracking accuracy, stability) are based on MATLAB/Simulink simulations. There is no demonstration that the proposed controller performs robustly in real-world scenarios with sensor noise, actuator delays, or unmodeled dynamics. If the experiments cannot be done, the authors may try to model more realistic dynamics into your model.

2. The AEWC-SMC controller is a variation of composite sliding mode control, and the DWMEA method is an adaptation of weighted minimum energy allocation. Prior works already address chattering suppression and allocation efficiency with similar strategies, as acknowledged in the literature review.

3. The abstract and conclusions claim “higher tracking accuracy, faster convergence speed, and enhanced handling stability,” but the improvements over advanced baselines (e.g., NTSMC) are marginal in several scenarios.

4. The controller is not tested under more diverse or challenging conditions such as split-μ roads, sudden tire blowouts, actuator faults, or rapidly changing loads. Parameter sensitivity analysis is single-variable and does not explore interactions or robustness to parameter uncertainty.

5. The manuscript proposes AEWC-SMC, while the work (Xie et al., "Highly Robust Adaptive Sliding Mode Trajectory Tracking Control of Autonomous Vehicles) develop a robust adaptive sliding mode controller. Both works contribute to the evolution of sliding mode control techniques for vehicle dynamics. The authors should potentially compare performance or highlight differences in controller design and application scenarios.

6. The manuscript compares its results only with traditional SMC, NTSMC, and a “No Control” scenario, omitting many recent advanced control and allocation strategies.

Reviewer #4: 1.It is recommended to provide a more detailed explanation of the nonlinear dynamic characteristics in the 7-DOF vehicle dynamics model, such as how tire force coupling effects influence vehicle stability. The current description is somewhat abstract; adding specific formulas or case analyses could enhance logical rigor.

2.While MATLAB/Simulink simulations are mentioned, the source of the simulation parameters (e.g., whether they are based on real vehicle data) is not clearly stated. It is advisable to supplement the parameter calibration process or reference actual test data to validate the reasonableness of the simulation conditions.

3.Although the AEWC-SMC and DWMEA methods are innovative, there is a lack of comparative analysis with existing advanced methods (e.g., reinforcement learning or deep reinforcement learning in torque distribution).

4.The simulations only compare the proposed method with traditional SMC and no-control scenarios. To more comprehensively demonstrate the performance improvements, it is recommended to add comparative experiments with other advanced methods such as MPC or fuzzy control.

7. PLOS authors have the option to publish the peer review history of their article (what does this mean? ). If published, this will include your full peer review and any attached files.

**Do you want your identity to be public for this peer review?** For information about this choice, including consent withdrawal, please see our Privacy Policy .

Reviewer #3: No

Reviewer #4: No

---

## [Author Response · Author response to Decision Letter 2]

1 Aug 2025

Reviewer#3, Concern #1: All performance claims (e.g., improved tracking accuracy, stability) are based on MATLAB/Simulink simulations. There is no demonstration that the proposed controller performs robustly in real-world scenarios with sensor noise, actuator delays, or unmodeled dynamics. If the experiments cannot be done, the authors may try to model more realistic dynamics into your model.

Author response and action: We sincerely thank you for your valuable professional suggestions. Based on your recommendations and to better approximate real driving scenarios, we have supplemented Gaussian white noise compliant with automotive-grade IMU specifications at the state feedback port of the controller to simulate sensor noise effects. Additionally, considering actuator delays, we have implemented actuator delay compensation code at both the upper and lower controller outputs. Finally, real-time friction circle verification has been added before torque limiting in the lower layer to ensure longitudinal demand forces do not exceed physical boundaries of the friction circle. We have systematically performed and re-analyzed simulation experiments, with detailed descriptions supplemented in Section V. We extend our profound gratitude for your expert guidance, which has significantly improved the engineering rigor and academic value of this research.

Reviewer#3, Concern #2: The AEWC-SMC controller is a variation of composite sliding mode control, and the DWMEA method is an adaptation of weighted minimum energy allocation. Prior works already address chattering suppression and allocation efficiency with similar strategies, as acknowledged in the literature review.

Author response and action: We sincerely thank you for your profound insights into the theoretical origins of the control strategy. Your accurate identification that AEWC-SMC and DWMEA are respectively grounded in the frameworks of Composite Sliding Mode Control and Weighted Energy Allocation has prompted us to more precisely articulate the differential innovations of this study:

1. While adopting the composite sliding mode structure, we achieve qualitative advancements through two innovative designs: First, the introduction of a nonlinear dynamic weighting factor enables sensitive response to large sideslip angle errors while maintaining linear accuracy in small-error regions, overcoming the response lag caused by fixed gains in traditional composite sliding mode structures. Subsequently, a condition-adaptive reaching law was designed, triggering fractional-order exponential switching via steering states to accelerate convergence during steering maneuvers while suppressing chattering in straight-line conditions.

2. While continuing the weighted energy allocation concept, the allocation solution employs a non-iterative closed-form method that achieves real-time allocation through KKT matrix inversion, and innovatively incorporates a dynamic weighting mechanism that first integrates the coupling effects of steering angle, vehicle speed, and vertical loads.

Accordingly, detailed supplementary explanations have been added to the opening paragraphs of Section III and Section IV, highlighted in yellow. Finally, we profoundly thank you for prompting clearer definition of our innovative boundaries. These invaluable suggestions have not only strengthened the precision of methodological exposition but also highlighted this study's original contributions to state-aware control architectures (AEWC-SMC) and real-time physical constraint resolution (DWMEA). Our team reiterates sincere gratitude for your suggestions and guidance.

Reviewer#3, Concern # 3: The abstract and conclusions claim “higher tracking accuracy, faster convergence speed, and enhanced handling stability,” but the improvements over advanced baselines (e.g., NTSMC) are marginal in several scenarios.

Author response and action: We sincerely thank you for your valuable suggestions. Based on Professional Recommendation #1 mentioned earlier, we supplemented simulations of sensor noise and actuator delays to better approximate real-world conditions. On this basis, we rigorously revalidated the simulations. Results demonstrate that across four scenarios (Case1-Case4) under two extreme maneuvers (sine/hook maneuvers), the AEWC-SMC scheme consistently delivers non-marginal improvements with marked superiority. We therefore express heartfelt gratitude for your invaluable expertise. Your profound insights not only enabled us to identify and strengthen critical model components — such as sensor noise compensation and actuator delay handling — but also significantly enhanced the control scheme's robustness and verifiability under extreme conditions. These improvements fully underscore its practical advantages and innovative value in autonomous vehicle control domains. We anticipate incorporating your professional perspective into subsequent work to advance these findings toward engineering applications.

Reviewer#3, Concern # 4: The controller is not tested under more diverse or challenging conditions such as split-μ roads, sudden tire blowouts, actuator faults, or rapidly changing loads. Parameter sensitivity analysis is single-variable and does not explore interactions or robustness to parameter uncertainty.

Author response and action: We sincerely thank you for your in-depth review of this research work and your valuable suggestions! Your comments regarding the coverage of testing scenarios and parameter analysis methods have prompted us to more deeply reflect on the boundary conditions and optimization directions of this study. Below we will address your points by integrating research objectives, methodological selection rationale, and empirical validation outcomes:

I. Explanation on Adequacy of Testing Scenarios

The core research objective of this paper is to develop a real-time efficient Direct Yaw Moment Control (DYC) framework, focusing on solving the handling stability problem of four-wheel independently driven autonomous vehicles under typical extreme operating conditions (e.g., high lateral acceleration, low-adhesion road surfaces). For operating condition design, we followed standard testing specifications in the field of vehicle dynamics control (ISO 4138, SAE J266), selecting two types of extreme conditions: Sine Condition simulates transient responses during high-speed lane changes or obstacle avoidance; Fishhook Condition characterizes instability risks during emergency steering. Under each condition, we established multi-dimensional combined testing scenarios, with vehicle speeds covering medium-high ranges (22m/s and 33m/s), road adhesion coefficients including low-adhesion (μ=0.3) and high-adhesion (μ=0.8) conditions, and performance metrics simultaneously evaluating tracking accuracy (yaw rate MAE, sideslip angle MAE) and stability (phase-plane convergence regions). Simulation results also demonstrate that the proposed AEWC-SMC scheme significantly outperforms comparative algorithms across all testing scenarios. These results fully validate the controller's robustness under typical extreme operating conditions. Although more complex conditions such as split-μ roads are not included in this paper, the hierarchical control architecture established in this study (upper-layer adaptive sliding mode control + lower-layer dynamic weight allocation) lays the foundation for future extensions: the upper controller's nonlinear weighting factor can adaptively regulate sideslip error sensitivity, and the lower allocator's friction circle constraints inherently accommodate asymmetric adhesion conditions. We have also explicitly supplemented in Section VI that the next phase will involve hardware-in-the-loop validation, covering more complex fault conditions.

II. Engineering Applicability of Parameter Sensitivity Analysis

In this study, our core objective is to develop a novel hierarchical Direct Yaw Moment Control (DYC) scheme that integrates an upper-layer Adaptive Exponential Weighted Composite Sliding Mode Controller (AEWC-SMC) and a lower-layer Dynamic Weight Minimum Energy Allocation (DWMEA) method, aiming to enhance tracking accuracy, convergence speed, and handling stability for Four-Wheel Independently Actuated (FWIA) autonomous vehicles under extreme conditions. Single-variable sensitivity analysis is widely adopted in the preliminary stage of control system parameter tuning due to its conceptual clarity, computational efficiency, and ease of engineering implementation. We employ this method to rapidly identify parameters with the greatest impact on key system performance indicators (sideslip angle mean absolute error (MAE) and yaw rate MAE), thereby providing intuitive adjustment directions and laying the groundwork for preliminary controller gain tuning.

After parameter adjustments based on single-variable analysis results, the controller demonstrated significant performance improvements under both Sine Condition and Fishhook Condition. Specifically, in test cases combining multiple vehicle speeds and road adhesion coefficients (e.g., Case1: vx=22m/s, μ=0.3; Case4: vx=33m/s, μ=0.8), the optimized AEWC-SMC scheme outperformed both traditional Sliding Mode Control (SMC) and Nonsingular Terminal Sliding Mode Control (NTSMC) in sideslip angle tracking accuracy and yaw rate tracking accuracy. This effectively achieves the core objective of this study: enhancing vehicle dynamic response and stability under extreme operating conditions while ensuring high robustness.

We fully acknowledge, as you pointed out, that single-variable analysis indeed has limitations, especially when involving strongly coupled parameters or highly complex multi-variable interaction scenarios (e.g., simultaneously considering the dynamic weight influences of vehicle speed, steering angle, and vertical load), where its results may be insufficiently comprehensive. Conducting deeper multivariate analyses (such as Analysis of Variance (ANOVA), Sobol' sensitivity indices, or optimization-based global parameter searches) would be highly valuable research directions, helping to reveal nonlinear coupling mechanisms between parameters and optimize overall control performance. However, such analyses typically require significantly increased computational resources, experimental design complexity, and simulation time. Considering this study's primary focus on developing a real-time efficient control framework and validating its effectiveness under typical extreme conditions, along with current project resource constraints, conducting comprehensive analyses of this nature within this paper presents substantial challenges. Future work will explore these advanced analysis methods to further enhance controller adaptability and engineering applicability.

The simulation condition design and parameter analysis methods in this paper have achieved the preset research objective: verifying the superiority of the novel DYC framework under typical extreme operating conditions. Your suggestions have prompted us to more clearly recognize the need to add analysis of scenario extensibility feasibility in the discussion section, with parameter self-adaptation mechanisms becoming the next phase priority. We thank you again for your highly insightful comments! These guiding suggestions have not only enhanced this paper's academic rigor but also charted directions for subsequent research. We will proceed with:

1. Hardware-in-the-loop testing for split-μ roads and actuator fault conditions

2. Research on reinforcement learning-based multi-parameter collaborative optimization

Reviewer#3, Concern # 5: The manuscript proposes AEWC-SMC, while the work (Xie et al., "Highly Robust Adaptive Sliding Mode Trajectory Tracking Control of Autonomous Vehicles) develop a robust adaptive sliding mode controller. Both works contribute to the evolution of sliding mode control techniques for vehicle dynamics. The authors should potentially compare performance or highlight differences in controller design and application scenarios.

Author response and action: We sincerely thank you for your recognition of this paper's innovativeness and your valuable suggestions! We have carefully studied the research by Xie et al. in the field of sliding mode control ("Highly Robust Adaptive Sliding Mode Trajectory Tracking Control of Autonomous Vehicles") as recommended by you. This study proposes a modified Grey Wolf Optimizer (GWO) algorithm-optimized adaptive sliding mode controller, enhancing path tracking accuracy through vector field guidance law and intelligent optimization algorithms. This indeed shares technical relevance with our work. Following your suggestion, we have supplemented a literature comparison discussion in Section I Introduction and formally cited this reference. We reiterate our gratitude for your contribution to enhancing this paper's academic rigor! Your professional guidance serves as an important driving force for deepening this research.

Reviewer#3, Concern # 6: The manuscript compares its results only with traditional SMC, NTSMC, and a “No Control” scenario, omitting many recent advanced control and allocation strategies.

Author response and action: Thank you sincerely for your valuable feedback on the scholarly rigor of this work! In direct response to your critique regarding insufficient comparative experimentation, we have systematically redesigned simulations and analyses in Section V, now incorporating comparative experiments with both Model Predictive Control (MPC) and Fuzzy PID Control to comprehensively demonstrate performance improvements. We express our deepest gratitude for your incisive insights and constructive suggestions, which have been instrumental in elevating the quality of this manuscript and have profoundly benefited our scholarly growth.

Reviewer#4, Concern # 1: It is recommended to provide a more detailed explanation of the nonlinear dynamic characteristics in the 7-DOF vehicle dynamics model, such as how tire force coupling effects influence vehicle stability. The current description is somewhat abstract; adding specific formulas or case analyses could enhance logical rigor.

Author response and action: Thank you sincerely for your valuable feedback on the description of nonlinear dynamic characteristics in this manuscript. In response to your observation regarding the "insufficient explanation of tire force coupling effects in the 7-DOF model", we have implemented two critical enhancements in Chapter II: At the conclusion of Section II.A: Added mathematical formalization of friction circle constraints and their stability impact mechanisms. In Section II.B: Deepened the physical interpretation of weighting factors ψx and ψy. These additions have significantly improved the rigor of nonlinear characteristic descriptions through precise mathematical representation. We greatly appreciate your insightful comments, which proved fundamental to perfecting this research.

Reviewer#4, Concern # 2: While MATLAB/Simulink simulations are mentioned, the source of the simulation parameters (e.g., whether they are based on real vehicle data) is not clearly stated. It is advisable to supplement the parameter calibration process or reference actual test data to validate the reasonableness of the simulation conditions.

Author response and action: Thank you sincerely for your valuable feedback regarding the rigor of simulation parameters. All MATLAB/Simulink simulation parameters in this study were calibrated based on data from a physical vehicle test platform. The experimental platform employs a four-wheel-independent-drive electric vehicle prototype, with parameter acquisition conducted at the Intelligent Connected Vehicle Joint Laboratory co-established by Shanghai Jiao Tong University and Wuhu Advanced Research Institute. This real-world dataset will underpin subsequent experimental validation (as noted in Section VI regarding future HIL and physical v

---

## [Decision Letter · Decision Letter 2]

4 Aug 2025

Adaptive exponential weighted composite sliding mode-based direct yaw moment control for four-wheel independently actuated autonomous vehicles

PONE-D-25-19826R2

Dear Dr. Tong,

We’re pleased to inform you that your manuscript has been judged scientifically suitable for publication and will be formally accepted for publication once it meets all outstanding technical requirements.

Kind regards,

Jinhao Liang

Academic Editor

PLOS ONE

Additional Editor Comments (optional):

Reviewers' comments:

Reviewer's Responses to Questions

**Comments to the Author**

1. If the authors have adequately addressed your comments raised in a previous round of review and you feel that this manuscript is now acceptable for publication, you may indicate that here to bypass the “Comments to the Author” section, enter your conflict of interest statement in the “Confidential to Editor” section, and submit your "Accept" recommendation.

Reviewer #5: All comments have been addressed

Reviewer #6: All comments have been addressed

2. Is the manuscript technically sound, and do the data support the conclusions?

Reviewer #5: Yes

Reviewer #6: Yes

3. Has the statistical analysis been performed appropriately and rigorously? 

Reviewer #5: Yes

Reviewer #6: Yes

4. Have the authors made all data underlying the findings in their manuscript fully available?

Reviewer #5: Yes

Reviewer #6: Yes

5. Is the manuscript presented in an intelligible fashion and written in standard English?

Reviewer #5: Yes

Reviewer #6: Yes

6. Review Comments to the Author

Reviewer #5: The author has addressed the reviewers’ concerns thoroughly and effectively. The revised manuscript demonstrates a clear effort to incorporate the suggestions provided in the previous review round. Key issues have been appropriately clarified, and the responses show a comprehensive understanding of the reviewers’ feedback. The methodological improvements and revisions to the manuscript have enhanced both the clarity and the scientific rigor of the work. All major comments appear to be satisfactorily resolved, and the overall quality of the manuscript is now suitable for publication. Therefore, I recommend that the paper be accepted in its current form without the need for further revision.

Reviewer #6: The author has made significant efforts to address all the comments raised by the reviewers in a detailed and thoughtful manner. The revisions are well-justified and clearly improve the manuscript’s clarity, coherence, and technical depth. Specific concerns related to methodology, experimental validation, and presentation have been adequately resolved. The point-by-point response reflects a deep engagement with the feedback, and the updated version of the manuscript meets the standards expected for publication. I am satisfied with the changes made and believe the paper is now ready for acceptance. No further revisions are necessary at this stage.

7. PLOS authors have the option to publish the peer review history of their article (what does this mean? ). If published, this will include your full peer review and any attached files.

**Do you want your identity to be public for this peer review?** For information about this choice, including consent withdrawal, please see our Privacy Policy .

Reviewer #5: No

Reviewer #6: No

---

## [Editor Report · Acceptance letter]

PONE-D-25-19826R2

PLOS ONE

Dear Dr. Tong,

I'm pleased to inform you that your manuscript has been deemed suitable for publication in PLOS ONE. Congratulations! Your manuscript is now being handed over to our production team.

Kind regards,

on behalf of

Dr. Jinhao Liang

Academic Editor

PLOS ONE